# Reconstruction of stochastic temporal networks through diffusive arrival times

Xun Li[1] & Xiang Li[1]

Temporal networks have opened a new dimension in defining and quantification of complex interacting systems. Our ability to identify and reproduce time-resolved interaction patterns is, however, limited by the restricted access to empirical individual-level data. Here we propose an inverse modelling method based on first-arrival observations of the diffusion process taking place on temporal networks. We describe an efficient coordinate-ascent implementation for inferring stochastic temporal networks that builds in particular but not exclusively on the null model assumption of mutually independent interaction sequences at the dyadic level. The results of benchmark tests applied on both synthesized and empirical network data sets confirm the validity of our algorithm, showing the feasibility of statistically accurate inference of temporal networks only from moderate-sized samples of diffusion cascades. Our approach provides an effective and flexible scheme for the temporally augmented inverse problems of network reconstruction and has potential in a broad variety of applications.

[1] Adaptive Networks and Control Laboratory, Department of Electronic Engineering, and Research Center of Smart Networks and Systems, School of Information Science and Engineering, Fudan University, Shanghai 200433, China. Correspondence and requests for materials should be addressed to Xiang Li (email: lix@fudan.edu.cn).

The collective functionality of complex networks emerges as a consequence of the interactions among their constituents. Recently, it has been empirically observed that, besides the spatially or topologically structured organization, the temporal constraints imposed on many complex interacting systems add a further dimension crucial for understanding their generic structure and dynamics. Internal links in these so-called temporal networks[1] evolve over time and are inherently registered by a series of rhythmically activated events among interacting actors at discrete time stamps. The causal sequence of the time-ordered links strikingly affects accessibility[2], an essential and revealing characteristic, especially in social and communication networks related to human activities. Such a profound shift from static to temporally intermittent interactivity properties has expanded, challenged or even redefined many fundamental concepts of networks, including path length[3,4], clustering correlation[3,4], node centrality[3,5], communicability[6], structural controllability[7], motif[8] and community structure[9].

Of another interest are the anomalous patterns (for example, bursts and heavy tails) embedded in the interevent time (IET) distribution of temporal interactions[10,11]. This marked departure of human activities from Poissonian behaviour substantially alters the dynamical process that takes place on networks, such as epidemic spreading[10–14], random walks[15,16], synchronization[17], cooperative evolution[18], consensus and coordination processes[19,20]. Undoubtedly, identifying the temporal interaction pattern is a first step in understanding and controlling collective dynamics of empirical temporal networked systems.

The vast majority of generative models for temporal networks—aimed at reproducing the empirical sequences of the time-stamped interactions—by and large require a priori knowledge of the raw data, or relevant statistics on the underlying interaction patterns[21–24]. However, strong limitations often arise regarding the availability of time-resolved interaction data at the individual level. Apart from increased technological expenditures, data collection is also hindered by small observation windows or samples reflecting individual activities only at coarse time granularity. Some data sets potentially suffer from other statistical deficiencies resulting from, for example, the participants with highly correlated behaviour from a skewed population. Another, even more problematic, restriction is imposed by data observability issues. The specifics of temporal interactions, particularly in social and financial networks, are in general obscured due to the privacy concerns of the participants, making their time-extended network structure unobservable, at least in principle. Extracting temporal networks from measurable data at the collective level has hence become a very desirable task. In viral marketing, for example, mining social networks is of critical value to identify highly influential customers who directly affect other consumers' decision making[25]. However, the interpersonal interactions (word-of-mouth recommendations) are often rendered private and thus indirectly accessible. In such circumstances, the recorded product dissemination history (for example, when customers purchase the products) provides an observable data source for the social network mining. Other candidate data applicable to similar inferential tasks include observed information cascades, such as propagating memes through blog posts[26] and tweets[27] on social media.

In this paper, we focus on exploiting such time-of-arrival data collected from the diffusion process taking place on networks, partially because diverse temporal interactions serve as the local propagation mechanisms for material or information exchange across a population in a variety of realistic scenarios, ranging from the spread of infectious diseases to the diffusion of cultural fads and the proliferation of innovative ideas (see, for example, a recent review[28]). We show later on that it is fundamentally possible to learn latent networks by discovering both structural and temporal regularities in diffusion process data.

Here we restrict on reconstructing a class of stochastic temporal networks (STNs)—which can be generally taken as a null model preserving temporal statistics for all dyads of interacting individuals but ignoring higher-order correlations across them[15]. Specifically, an STN builds, on the basis of a time-aggregated static network, an extra temporal dimension by assigning to each link a mutually independent random IET, denoting the interval of activation of the events occurring on the link in a renewal manner[29]. This simplification was at first made for analytical convenience[15], and a number of exact methods have in recent years been developed for quantifying diffusion dynamics of complex networked systems when temporal characteristics of pairwise interactions are incorporated[16,28,30,31]. Compared to purely phenomenological (for example, regression-based) models[32–34], the STN is endowed with better predictive power in both theoretical and applied domains (see Supplementary Note 1 for a brief review on the related literature). Towards an effective null modelling procedure for temporal networks in a data-driven fashion, we conceive the STN model as a convenient descriptive device for explaining time-course data of observed diffusion processes thereon, and we carry out extensive benchmark tests on a variety of simulated and empirical temporal networks to validate the reconstruction efficacy of our approach. We further discuss the inferential complexity of temporal networks in terms of entropy of underlying diffusion pathways from an information-theoretic viewpoint.

## Results

**Overall sketch of the reconstruction method.** The null modelling method for temporal networks we develop is to some extent intuitive and pragmatic. As sketched in Fig. 1, the topological structure of underlying temporal networks can be directly recovered using the superposed spreading paths encoded in the observable arrival order in the diffusion processes. On the other hand, the statistical temporal properties of dyadic interactions can be also exacted from the time differences of arrivals in time courses of diffusion, an incomplete observation of waiting times associated with possible diffusion routes. We show that in both cases, a soft (namely, probabilistic) censoring indicator whether a link lies in actual diffusion routes, called branching coefficient, plays a pivotal role in our inferential framework. In the following, we first introduce the forward model for information diffusion on temporal networks, and describe the construction of first-order STNs by decorrelating the dyadic interaction sequences of an empirical network. We then derive the likelihood of observing a specific diffusion cascade as well as the corresponding branching coefficients. Based on them, we exhibit a coordinate-ascent scheme which alternates between estimating the latent time-aggregated network by Markov chain Monte Carlo (MCMC) techniques and determining the dyad-level WTDs from self-consistency conditions. Additional details on mathematical proofs, performance assessments of algorithms, empirical validation results, as well as further discussions are deferred in Supplementary Notes 2–12.

**Forward generative model.** We first introduce the forward model used to describe data-generating processes. The temporal network $\mathcal{N} = (\mathcal{V}, \mathcal{E})$ on which diffusion takes place is represented in terms of a set of nodes $v \in \mathcal{V}$ as well as a set of events $(u, v, t, \delta t) \in \mathcal{E}$ observed within a time window $[0, T^{\mathrm{w}}]$. Each event is depicted by a temporal link from node $u$ to node $v$ activated during a time interval $[t, t + \delta t)$. Here, we assume for simplicity that the duration of events is infinitesimally small

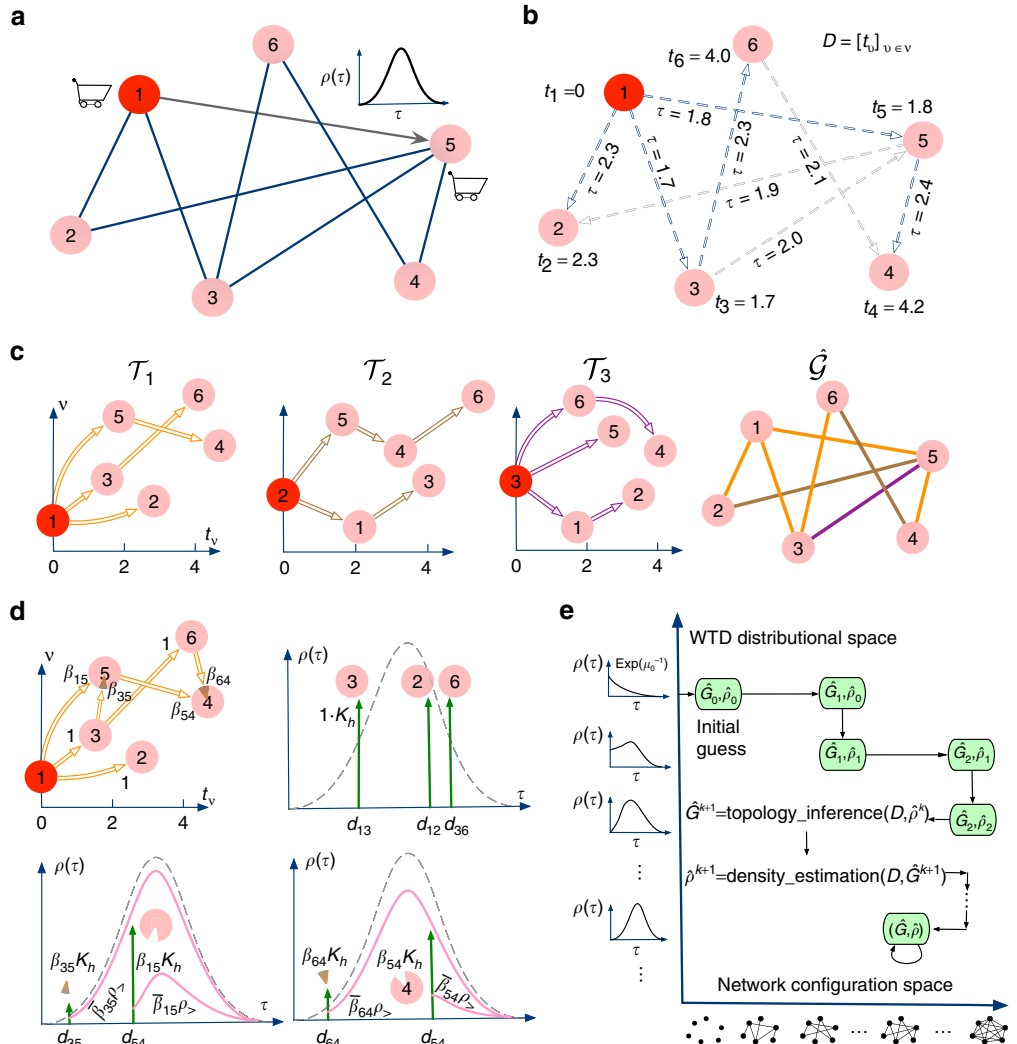

**Figure 1 | Schematic representation of our inferential framework.** (**a**) The STN tuple $\mathcal{N}_s = (\mathcal{G}, \rho)$ to be inferred which confines the underlying diffusion process, both spatially and temporally. Specifically, a diffusion cascade originated by any source node (red circle) can only proceed on the time-aggregated graph $\mathcal{G}$, and relay delay of diffusion along any network link (green arrow) is a random variable $\tau$ (called waiting time) with the corresponding distribution denoted by WTD $\rho(\tau)$. (**b**) A simulated diffusion cascade using our generative model with realized waiting times attached to network links. Here, DAT data $\mathcal{D} = [t_v]_{v \in \mathcal{V}}$ are obtained from first-arrival observations of the diffusion cascade, with each entry $t_v$ equal to the length of the shortest diffusion path (blue dashed arrow) connecting nodes $s^*$ and $v$. The dashed arrows highlight that the waiting times are unobservable in our problem settings. (**c**) An illustrative result of topology inference using multiple diffusion cascades. Intuitively, the network topology $\mathcal{G}$ can be recovered by uniting the realized (or inferred) diffusion paths, $\mathcal{G} \leftarrow \bigcup_{i=1}^{C} \mathcal{T}_i$ ($\hat{\mathcal{G}} \leftarrow \bigcup_{i=1}^{C} \hat{\mathcal{T}}_i$). Here, $\mathcal{T}_i$ denotes the underlying diffusion tree associated with the observed cascade, which is inferable from diffusion data as a spanning tree $\hat{\mathcal{T}}_i$ with maximum link weights $\beta_{uv}$ (called branching coefficient, see definition below). (**d**) Kernel density estimation for WTD $\rho(\tau)$ using $\mathcal{D} = [t_v]$ as incomplete observations of underlying waiting times. For brevity, we adopt Dirac delta kernel function $K_h(\tau) = \delta(\tau)$. The sector located at each node $v$ represents its associated branching coefficients, each $\beta_{uv}$ defining the probability that an upstream neighbouring node $u \in \mathcal{I}_v$ spreads the information to node $v$ in a specific cascade. Thus the self-consistent estimator $\hat{\rho}(\tau)$ simply sums over all observed cascades (only one is shown here) Dirac deltas of amplitude $\beta_{uv}$ placed at the TDOAs, $d_{uv} = t_v - t_u$, as well as $\hat{\rho}(\tau)$ itself truncated above $d_{uv}$ and then renormalized to $\bar{\beta}_{uv} = 1 - \beta_{uv}$. The first term corresponds to the contribution from link $(u,v)$ as a diffusion branch leading to the tight constraint $\tau_{uv} = d_{uv}$ with probability $\beta_{uv}$, and the second from link $(u,v)$ as a chord having the censored waiting time $\tau_{uv} > d_{uv}$ with the complementary probability $\bar{\beta}_{uv}$. (**e**) The coordinate-ascent implementation of our inferential method. The overall procedure is initialized with a first guess $\hat{\rho}_0$ for the WTD, and alternates until convergence between the topology inference and density estimation steps using observed diffusion data and the most recent estimates. Additional algorithmic details can be found in Supplementary Note 7.

($\delta t \to 0$) in order to avoid several links to be present simultaneously. Accordingly, the STN model $\mathcal{N}_s$ built on $\mathcal{N}$ consists of two key ingredients. First, we obtain the time-aggregated graph $\mathcal{G}$ by projecting the temporal events $(u, v, t) \in \mathcal{E}$ onto topological links $(u, v) \in \mathcal{G}$, each having as its pre-image a list of events, $\{(u, v, t_{uv}^1), (u, v, t_{uv}^2), \ldots\}$, in order of their activation times. Second, to generate statistically accurate temporal links, we substitute random sequences of synthesized events with IETs drawn

i.i.d. from the probability density function $\psi_{uv}(t)$ fitted on the empirical data $\{t_{uv}^{i+1} - t_{uv}^i\}$ for each link separately. It is noteworthy here that the STN model defines ensembles of random temporal networks with the realistic IET distributions at the dyadic level, while being simplified by the automatic elimination of inter-link correlations.

Consider now a node $s^* \in \mathcal{V}$ introduced as the information source that initializes the diffusion process on $\mathcal{N}$, during which

each node has one of two mutually exclusive states: (i) informed, if it has already received the information through any incident link; or (ii) ignorant, if it has not been informed so far. In a broadcast manner, the information is transmitted along any time-respecting path which consists of consecutive events of increasing times [that is, $\{(s^*, v_1, t_1), (v_1, v_2, t_2), \ldots, (v_{l-1}, v_l, t_l)\} \subset \mathcal{E}$ with $t_1 < \ldots < t_l$]. The diffusive arrival time (DAT) at node $v$ is defined as the instant at which node $v$ enters the informed state, and the path through which the information arrives first is termed the fastest time-respecting path (FTP). Here, we assume without loss of generality that $t_v < \infty$ for any node $v \in \mathcal{V}$; in other words, the underlying network is connected and contains at least one time-respecting path between any source and receiver pair.

**Likelihood functional for the observed cascade.** We next calculate the likelihood of observing a single diffusion cascade of DATs $\mathcal{D} = [t_v]_{v \in \mathcal{V}}$, attempting to search the STN model $\mathcal{N}_s$ that fits to the diffusion data. To this end, it is conventional to introduce the concept of the waiting time $\tau_{uv}$ occurring on each link $(u, v) \in \mathcal{G}$. It is defined as the relay interval for which node $u$ has to wait since it becomes informed until it activates a next link incident to node $v$ for transmitting the information. In the case of uncorrelated IETs, $\tau_{uv}$ is randomly drawn according to a length-biased probability law which takes the form[35]

$$\rho_{uv}(\tau) = m_{uv}^{-1} \int_\tau^\infty \psi_{uv}(t) \mathrm{d}t \Theta(\tau), \tag{1}$$

where $m_{uv} = \int_0^\infty t \psi_{uv}(t) \mathrm{d}t$ is the mean of the IET distribution $\psi_{uv}(t)$, and $\Theta(\cdot)$ is the Heaviside unit-step function. Denote $\boldsymbol{\rho} = [\rho_{uv}(\tau)]_{(u,v) \in \mathcal{G}}$ by the set of such WTDs assigned to respective links of $\mathcal{G}$, and the STN model is thus also fully described by $\mathcal{N}_s = (\mathcal{G}, \boldsymbol{\rho})$, as illustrated in Fig. 1a. Hence the likelihood of observing a given cascade $\mathcal{D}$ considers all possible FTPs weighted by their conditional probabilities of occurrence and is written

$$L(\mathcal{D}|\mathcal{N}_s) = \sum_{\mathcal{T} \subset \mathcal{G}} L(\mathcal{D}, \mathcal{T}|\mathcal{N}_s)$$

$$= \sum_{\mathcal{T} \subset \mathcal{G}} \prod_{(u,v) \in \mathcal{T}} \rho_{uv}(d_{uv}) \prod_{(u,v) \notin \mathcal{T}} \int_{d_{uv}}^\infty \rho_{uv}(\tau) \mathrm{d}\tau, \tag{2}$$

where $L(\mathcal{D}, \mathcal{T}|\mathcal{N}_s)$ is the likelihood that the cascade $\mathcal{D}$ is observed along a set $\mathcal{T}$ of underlying FTPs, and $d_{uv} = t_v - t_u$ represents the time differences of arrival between nodes $u$ and $v$. Here $\mathcal{T}$ denotes any possible union of FTPs from source $s^*$ to respective other nodes compatible with the diffusion cascade $\mathcal{D}$. Note that $\mathcal{T}$ constitutes an acyclic directed subgraph of $\mathcal{G}$ with likelihood $1 - O(\delta t)$, and ideally, we call $\mathcal{T}$ an ($s^*$-rooted) diffusion tree. Obviously, the condition of $\mathcal{T}$ coinciding with actual diffusion pathways requires each branch $(u, v) \in \mathcal{T}$ to produce an exact waiting time, $\tau_{uv} = d_{uv}$, as well as each chord $(u, v) \notin \mathcal{T}$ to produce instead a right-censored (namely, bounded from below) waiting time, $\tau_{uv} > d_{uv}$, in order to guarantee the minimum-time optimality of the diffusion that occurs along $\mathcal{T}$, as shown in Fig. 1b. After some algebra (see Supplementary Note 2 for details), we have the following logarithmic likelihood functional:

$$\ell(\mathcal{D}|\mathcal{N}_s)$$

$$= \sum_{v \in \mathcal{V} \setminus s^*} \log \sum_{(u,v) \in G} \lambda_{uv}(d_{uv}) + \sum_{(u,v) \in G} \log \Phi_{uv}(d_{uv}), \tag{3}$$

where $\Phi_{uv}(\tau) = \int_\tau^\infty \rho_{uv}(t) \mathrm{d}t$ and $\lambda_{uv}(\tau) = \rho_{uv}(\tau)/\Phi_{uv}(\tau)$ are the survival and hazard functions of WTDs[36], respectively.

**Route-specific branching coefficient.** We are now in the position to introduce the branching coefficient $\beta_{uv}(\mathcal{D}, \mathcal{N}_s)$, the

conditional probability of link $(u, v)$ acting as a diffusion branch (that is, $u$ is the first to inform $v$) given a specific cascade $\mathcal{D}$ on STN $\mathcal{N}_s$. By uncorrelated properties of dyadic network interactions, it is simply given by

$$\beta_{uv}(\mathcal{D}, \mathcal{N}_s) = \frac{\lambda_{uv}(d_{uv})}{\sum_{(w,v) \in \mathcal{G}} \lambda_{wv}(d_{wv})}. \tag{4}$$

Equation (4) has a clear interpretation in terms of superposition of inhomogeneous point processes[37]. The simultaneous exposure of $v$ to any informed neighbour $u$ constitutes a set of competing risk events, leading to a cumulative hazard rate $\Lambda_v(t) = \sum_{(u,v) \in \mathcal{G}} \lambda_{uv}(t - t_u)$ at which diffusion will occur at time $t$. Thus equation (4) immediately follows from Superposition Theorem by equating $t$ to the realized value of $v$'s DAT, $t_v$. More detailed derivation of $\beta_{uv}(\mathcal{D}, \mathcal{N}_s)$ is presented in Supplementary Note 2.

Note that to recover the underlying STN $\mathcal{N}_s$ requires observation of multiple diffusion cascades triggered from different regions of the network. Suppose we collect a sample of independent cascades $D = \{\mathcal{D}^i\}$ and aim at finding the optimal tuple $\hat{\mathcal{N}}_s = (\hat{\mathcal{G}}, \hat{\boldsymbol{\rho}})$ as the maximizer of the following log-likelihood

$$\ell(D|\mathcal{N}_s) = \sum_{i=1}^C \ell(\mathcal{D}^i|\mathcal{N}_s), \tag{5}$$

where $C$ denotes the number of observed diffusion cascades.

To be concrete, we resort to an iterative coordinate-ascent strategy to search $\hat{\mathcal{G}}$ and $\hat{\boldsymbol{\rho}}$ alternately. In what follows we specify the two steps outlined above and present numerical results obtained by applying to a variety of benchmark STNs and empirical temporal networks.

**Time-aggregated topology inference.** This step takes as input observed DATs $D$ as well as WTDs $\boldsymbol{\rho}$ that are known or estimated. However, finding the maximum likelihood (ML) time-aggregated graph $\hat{\mathcal{G}}(D, \boldsymbol{\rho})$ belongs to a wide class of submodular function optimization problems, which is in general computationally hard (see Supplementary Note 3).

Alternatively, rather than picking out a single ML estimate, we apply MCMC to integrate over all possible configurations $\hat{\mathcal{G}}$ using weights proportional to their likelihood values $L(D|\hat{\mathcal{G}}, \boldsymbol{\rho})$. Here we employ a Gibbs sampler[38] which starts from an initial graph $\hat{\mathcal{G}}^0$ and iteratively flips any link, say $(u, v)$, with acceptance probability $p_{uv} = [1 + exp(-\Delta_{uv}\ell(D|\hat{\mathcal{G}}, \boldsymbol{\rho}))]^{-1}$ one by one, where $\Delta_{uv}\ell(D|\hat{\mathcal{G}}, \boldsymbol{\rho})$ represents the marginal gain of the link flipping operation, that is,

$$\Delta_{uv}\ell(D|\hat{\mathcal{G}}, \boldsymbol{\rho}) = \begin{cases} \ell(D|\hat{\mathcal{G}} \cup uv, \boldsymbol{\rho}) - \ell(D|\hat{\mathcal{G}}, \boldsymbol{\rho}) & (u, v) \notin \hat{\mathcal{G}} \\ \ell(D|\hat{\mathcal{G}} \setminus uv, \boldsymbol{\rho}) - \ell(D|\hat{\mathcal{G}}, \boldsymbol{\rho}) & (u, v) \in \hat{\mathcal{G}}. \end{cases}$$

$$= \sum_{i=1}^C \left\{ \log[1 \pm \beta_{uv}(\mathcal{D}^i, \hat{\mathcal{G}}, \boldsymbol{\rho})] \pm \log \Phi_{uv}(d_{uv}^i) \right\}, \tag{6}$$

where $\beta_{uv}(\mathcal{D}^i, \hat{\mathcal{G}}, \boldsymbol{\rho})$ are the branching coefficients defined in equation (4) using the current configuration $\hat{\mathcal{G}}$. Here we adopt the shorthand notation $\hat{\mathcal{G}} \cup uv$ ($\hat{\mathcal{G}} \setminus uv$) for the network obtained by adding (removing) link $(u, v)$ from $\hat{\mathcal{G}}$. As schematically shown in Fig. 1c, the proposed procedure is inclined to sample links with large aggregate branching coefficients. This is roughly equivalent to reconstruction of the network topology using as building blocks the diffusion trees inferred from independent cascades (see Supplementary Note 2 for discussion on the role of branching coefficients in our inferential scheme).

Next we quantify the performance of our inference procedure. In view of the structural sparsity of many real-world networked systems, we preferably select a measurement index, called break-

even point (BEP), which strikes the optimal balance between precision and recall on true-positive links, as illustrated in Fig. 2. To test our approach, we carry out extensive numerical simulations for various types of benchmark time-aggregated networks and WTDs. We evaluate the attained BEP, as well as two other standard indices, the area under the receiver operating characteristic curve and the area under the precision-recall curve (AUPR) (see Supplementary Note 4). Because high inference accuracy can always be achieved, we report in Table 1 the minimum sample size for assuring at least 0.95 area under the receiver operating characteristic, area under the precision-recall curve and BEP, respectively, showing that universal high inference accuracy can be achieved from only a moderate sample size of diffusion observations. More detailed descriptions and complete results of benchmark tests for our inference algorithm can be found in Supplementary Table 5.

**Waiting-time distribution estimation.** This step takes as input observed DATs $D$ as well as an underlying time-aggregated network $\mathcal{G}$. To implement nonparametric estimation of WTD $\hat{\boldsymbol{\rho}}(D, \mathcal{G})$, the major difficulty stems from the indeterminacy of the diffusion trees and the censored nature of waiting times occurring on chord links during the diffusion cascades. Here, we adopt the 'Redistribute-to-the-Right' formulation[39] as an imputation scheme to tackle this censored data problem, by which we obtain the following self-consistent equations for estimating the WTDs:

$$\hat{\rho}_{uv}(\tau) = \frac{1}{C}\sum_{i=1}^{C}\left\{ \begin{array}{l} \beta_{uv}(\mathcal{D}^i, \mathcal{G}, \hat{\boldsymbol{\rho}})K_h(\tau - d_{uv}^i) \\ + \left[1 - \beta_{uv}(\mathcal{D}^i, \mathcal{G}, \hat{\boldsymbol{\rho}})\right]\frac{\hat{\rho}_{uv}(\tau)\Theta(\tau - d_{uv}^i)}{\hat{\Phi}_{uv}(d_{uv}^i)} \otimes K_h(\tau) \end{array}\right\}, \tag{7}$$

where $K_h(\cdot)$ is a kernel smoother with bandwidth $h$, $\Theta(\cdot)$ is the Heaviside unit-step function, and $\otimes$ denotes the convolution

operator. The first term in the braces corresponds to the probability of link $(u,v)$ acting as a diffusion branch, and the second term corresponds to that of link $(u,v)$ acting as a chord which contributes a truncated WTD of $\hat{\rho}_{uv}^k(\tau)$ above $d_{uv}^i$, as shown in Fig. 1d. Particularly, in the case of an underlying tree $\mathcal{T}$, $\hat{\rho}_{uv}(\tau)$ reduces to standard kernel density estimation[40] as $C^{-1}\sum_{i=1}^{C}K_h(\tau - d_{uv}^i)$. In the most general case of an arbitrary network $\mathcal{G}$, the consistency of the WTD estimator $\hat{\boldsymbol{\rho}}(D, \mathcal{G})$ is in essence guaranteed by our choice of weights $[\beta_{uv}(\mathcal{D}^i, \mathcal{G}, \hat{\boldsymbol{\rho}})]$ in soft assignments of respective links to diffusion branches in an expectation-maximization manner[41]. We numerically verify the consistency of the estimator for non-identically distributed WTDs attached to a small-size network, as illustrated in Fig. 3. A rigorous proof is presented in Supplementary Note 5.

When only limited amount of diffusion data are available for inference, the reconstruction accuracy undergoes a phase transition as the sample increases (see Supplementary Fig. 6). Hence we explore the minimum relative sample size for simultaneously successfully reconstructing both underlying networks and associated WTDs only from observed diffusion cascades, as displayed in Table 1. Somewhat surprisingly, the same reconstruction accuracy can generally be achieved using even less diffusion cascades in a considerable fraction of benchmark tests, comparing to the case where the underlying WTDs are explicitly known. This counter-intuition can be in part explained by the fact that our self-consistent WTD estimator provides an adaptive mechanism for the fitting of observed diffusion data. Specifically, through the process of inference, the estimator acquires subtle features of the empirical WTD $\hat{\boldsymbol{\rho}}$ in accordance with the particular realizations of waiting times, and hence outperforms the result given the true WTD $\boldsymbol{\rho}$ which, even if observed data truly follow it, differs more or less from $\hat{\boldsymbol{\rho}}$, especially when the sample size is small. We design a parametric bootstrap procedure to provide confidence bands to assess the variability in the estimated WTDs. Detailed implementation

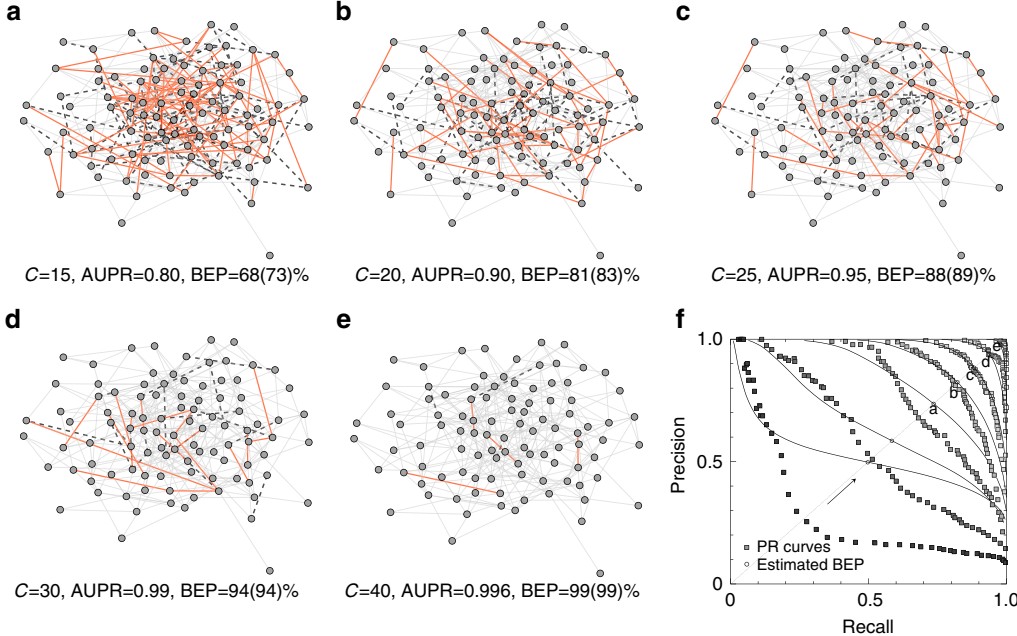

**Figure 2 | Break-even point of topology inference using diffusion cascades.** (**a–e**) The time-aggregated graph configuration with BEP optimality is attained by equating the number of false-positive links (red solid lines) to the number of false-negative links (black dashed lines) in estimated configurations. Percentages in parentheses correspond to the estimated BEPs. (**f**) Real and estimated PR curves with respect to increasing sample sizes of $C = 5, 10, 15, 20, 25, 30, 40$, respectively. The BEPs correspond to intersections of the PR curves with the diagonal of the unit square (dotted line). The time-aggregated network $\mathcal{G}$ is an Erdös–Rényi random graph of size $N = 100$ and average degree $\langle k \rangle = 6$, and the real WTD is Gaussian of mean $\mu = 2$ and variance $\sigma^2 = 0.4^2$, which is assumed here to be explicit for topology inference.

**Table 1 | Summary of critical relative sample sizes for STN reconstruction.**

| Networks | N | L | Gaussian | Weibull | Pareto |
|---|---|---|---|---|---|
| Lattice 2D | 100 | 180 | 0.03/0.07 | 0.03/0.08 | 0.04/0.08 |
| | | | 0.08/0.10 | 0.12/0.10 | 0.11/0.11 |
| Sierpinski | 123 | 243 | 0.04/0.09 | 0.02/0.07 | 0.02/0.05 |
| | | | 0.11/0.11 | 0.10/0.07 | 0.08/0.07 |
| Pseudofractal | 123 | 243 | 0.10/0.23 | 0.20/0.36 | 0.11/0.20 |
| | | | 0.48/0.24 | 0.58/0.59 | 0.27/0.38 |
| Apollonian | 124 | 366 | 0.13/0.29 | 0.24/0.49 | 0.18/0.33 |
| | | | 0.63/0.31 | 0.67/0.82 | 0.45/0.54 |
| Random graph | 100 | 291 | 0.08/0.14 | 0.16/0.29 | 0.16/0.26 |
| | | | 0.20/0.20 | 0.40/0.42 | 0.31/0.39 |
| Small-world | 100 | 300 | 0.05/0.10 | 0.08/0.19 | 0.10/0.18 |
| | | | 0.18/0.14 | 0.33/0.25 | 0.28/0.24 |
| Scale-free | 100 | 291 | 0.08/0.17 | 0.20/0.34 | 0.17/0.29 |
| | | | 0.24/0.24 | 0.49/0.49 | 0.35/0.45 |
| Karate | 34 | 78 | 0.15/0.30 | 0.31/0.66 | 0.30/0.47 |
| | | | 0.55/0.24 | 1.06/0.76 | 0.72/0.72 |
| Dolphins | 62 | 159 | 0.07/0.17 | 0.10/0.42 | 0.12/0.27 |
| | | | 0.36/0.15 | 0.55/0.40 | 0.43/0.31 |
| Miserables | 77 | 254 | 0.10/0.38 | 0.17/0.42 | 0.13/0.30 |
| | | | 0.73/0.24 | 1.18/0.58 | 0.55/0.39 |
| Football | 115 | 613 | 0.08/0.17 | 0.15/0.39 | 0.13/0.30 |
| | | | 0.27/0.20 | 0.73/0.51 | 0.46/0.40 |
| Jazz | 198 | 2,742 | 0.13/0.38 | 0.32/1.41 | 0.19/0.56 |
| | | | 0.81/0.31 | − /1.09 | 1.12/0.62 |

We apply our method to a variety of time-aggregated networks in combination with three types of benchmark WTDs, Gaussian, Weibull and Pareto distributions, to determine numerically the minimal relative sample size required for different reconstruction goals. More precisely, the (1,1)th-entry of each cell represents the critical value for assuring at least 0.95 AUROC, the (1,2)th-entry for 0.95 AUPR, the (2,1)th-entry for 0.95 BEP and the (2,2)th-entry (in blue) for simultaneous reconstruction of both underlying networks and WTDs, respectively. Here, $N$ is network size, $L$ is link number and relative sample size $C/N$ denotes the number of observed cascades relative to network size. Each critical relative sample size is an average over ten independent realizations and is obtained under the homogeneous population assumption, that is, all the STN links produce i.i.d. waiting times.

and illustrative numerical results are present in Supplementary Note 7.

We also apply our method to several realistic temporal networks to test its validity under more practical situations. Relative empirical validation results are reported in Table 2. Keeping in mind the gap between our STN model from reality, we introduce three statistical measures to quantify the deviation of empirical data sets from the null model assumption of i.i.d. distributed WTDs: distributional standard deviation, Pearson correlation coefficient and normalized mutual information. On one hand, we find that the WTDs underlying these real-world temporal networks have relatively small distributional standard deviation, which partially justifies our null model assumption. This considerable homogeneity in temporal interactivity of individuals can be interpreted by the typical distribution of human response time which is shown to be heavy-tailed with power exponent between 1 and 2. On the other hand, large Pearson correlation coefficient or normalized mutual information of empirical WTDs implies the salience of inter-link correlations, either (negatively) linear or non-linear, in many realistic temporal networks. This leads to substantially decreased reconstruction accuracy (see $BEP_1$ and $BEP_2$ in Table 2), suggesting correction of the self-consistent relation of WTDs to incorporate such polyadic correlations (see Supplementary Note 9 for discussion on the extended higher-order STN models). In addition to the discrepancy between empirical temporal networks and our STN model, another bottleneck causing the reduced $BEP_1$ comes from the iterative coordinate-ascent procedure. The reconstruction accuracy often collapses due to the positive feedback loop between topology inference and density estimation steps that amplifies the estimation error caused by the unrealistic null model assumptions. We further compare the reconstruction results using empirical WTDs as prior input to break such

unexpected feedback loops, showing an acceptable efficacy of our topology inference method even when applied to realistic temporal networks (see $BEP_3$ and $BEP_4$ in Table 2). We also find that the reconstruction of $\mathcal{G}$ can be sensibly improved given correct prior WTD $\rho$, and on the contrary, the estimation density for $\rho$ does not benefit from the addition of topological knowledge $\mathcal{G}$ (see Supplementary Figs 20–21). A possible reason lies in the extremely large supports of heavy-tailed WTDs, in which case our equidistant binning-based WTD estimator has a large number of parameters to be estimated from self-consistent iterations and hence demonstrates poor convergence properties (see Supplementary Note 10 for further discussion on the effect of binning and the Fourier-domain density estimation).

**Inferential complexity of temporal networks**. Numerical results of our benchmark tests reveal a complicated picture of how structural and temporal properties of the underlying network synergically affect the critical amount of diffusion data needed to attain a given reconstruction accuracy. To more comprehensively quantify the inferential complexity of diffusion structure embodied in temporal networks, we introduce a single entropic measure as follows:

$$\xi(\mathcal{D}, \mathcal{N}_s) = - \sum_{(u,v)\in\mathcal{G}} \beta_{uv}(\mathcal{D}, \mathcal{N}_s)\log\beta_{uv}(\mathcal{D}, \mathcal{N}_s). \quad (8)$$

Note that $\xi(\mathcal{D}, \mathcal{N}_s)$ reflects the entropy of possible diffusion trees associated with a given cascade $\mathcal{D}$ on $\mathcal{N}_s$, thus extending the concept of network complexity which counts the (logarithmic) number of spanning trees contained in a static graph[42]. Additional details on the derivation of equation (8) and related discussion are presented in Supplementary Note 11.

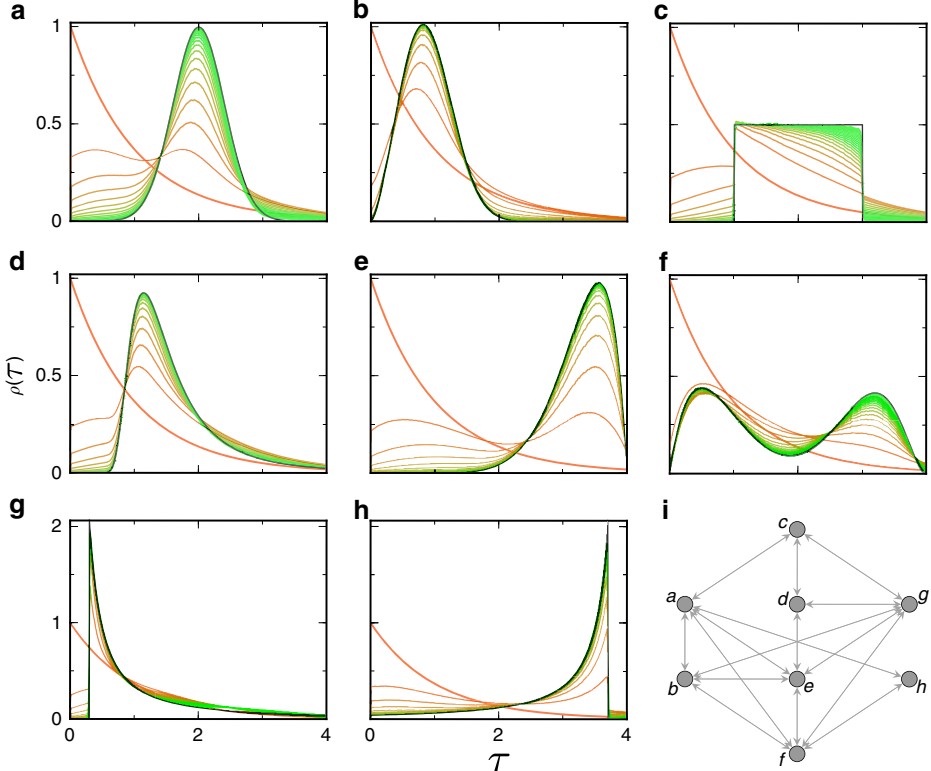

**Figure 3 | Asymptotic consistency of the iterative WTD estimator.** (**a**–**h**) Numerical verification is illustrated through large samples of diffusion cascades synthesized on a small-size network, with several benchmark WTDs (black lines) assigned to departure links of respective nodes. The iteratively estimated WTDs (gradient-coloured lines) ultimately converge to the correct distributions. (**i**) The illustrative network consisting of $N = 8$ nodes and $L = 16$ (double-directed) links. The sample size of diffusion cascades used for the WTD estimation is $C = 10^4$, the kernel bandwidth is chosen as $h = 10^{-3}$, and the *a priori* guess $\hat{\boldsymbol{\rho}}^0$ as exponentials of mean $\mu = 1$ (see Supplementary Note 6 for practical selection criteria of both $\hat{\boldsymbol{\rho}}^0$ and $h$).

**Table 2 | Empirical validation results using realistic temporal network data.**

| Data sets | $N$ | $L$ | DSD | PCC | NMI | $BEP_1$ | $BEP_2$ | $BEP_3$ | $BEP_4$ |
|---|---|---|---|---|---|---|---|---|---|
| Hospital | 75 | 32,424 | 0.076 | 0.021 | 0.516 | 0.25 | 0.98 | 0.70 | 0.76 |
| Workplace | 92 | 9,827 | 0.161 | −0.194 | 0.646 | 0.27 | 1.00 | 0.72 | 0.74 |
| HT09 | 113 | 20,818 | 0.109 | −0.234 | 0.441 | 0.20 | 0.96 | 0.63 | 0.71 |
| Primary | 242 | 125,773 | 0.086 | −0.280 | 0.164 | 0.36 | 0.97 | 0.76 | 0.80 |
| High2011 | 126 | 28,561 | 0.175 | −0.443 | 0.402 | 0.32 | 0.99 | 0.87 | 0.88 |
| High2012 | 180 | 45,047 | 0.113 | −0.206 | 0.487 | 0.33 | 1.00 | 0.88 | 0.88 |
| High2013 | 327 | 188,508 | 0.090 | −0.320 | 0.278 | 0.32 | 1.00 | 0.97 | 0.97 |

Here, $N$ is network size, $L$ is the number of temporal links, DSD, PCC and NMI denote the distributional s. d., Pearson correlation coefficient and normalized mutual information of empirical WTDs fitted to the data of empirical contact sequences, respectively (see Supplementary Note 8 for definitions). The inferential accuracies compared are the optimal precisions attained at break-even points of topology reconstruction using ($BEP_1$) empirical cascades realized on realistic temporal networks, ($BEP_2$) synthesized cascades on the STN model fitted to realistic temporal networks, ($BEP_3$) empirical cascades and empirical WTDs, and ($BEP_4$) synthesized cascades and empirical WTDs, respectively (see Supplementary Table 4 for detailed experimental settings). The relative sample size used for inference is set to $C/N = 1$. Relevant receiver operating characteristic and precision-recall curves for topology inference, as well as density estimation results for WTDs can be found in Supplementary Figs 20–21.

Next we examine the inferential complexity of diffusion structure in various types of benchmark WTDs, finding a natural positive correlation between the intrinsic complexity of the underlying diffusion processes and the critical relative sample size for reconstruction goals, as shown in Fig. 4. A more systematic analysis requires parametric generation of network ensembles that are tunable from both structural or temporal aspects to decouple the underlying influences. Here we first focus on three important structural properties of time-aggregated networks: clustering coefficient, average path length and connectivity heterogeneity. Numerical results show a nonlinear dependence of network complexity on the topological indices. Particularly in the practically relevant regime of salient small-world effects (characterized by large clustering coefficient and small average path length), the inferential

complexity is positively (negatively) correlated with clustering coefficient (average path length) of the time-aggregated network. For most specific distributional shapes used as our benchmark WTDs, the heterogeneity in degree distribution also increases the inferential complexity of networks. Intuitively, diffusion on small-world and/or highly heterogeneous networks is inclined to produce, due to frequent occurrences of competing-risk censoring, indiscernible 'downstream' DATs that are close to each other, thus complicating the task of network reconstruction. We further examine the roles of the first three (normalized central) moments of WTDs, finding that the inferential complexity is positively (negatively) correlated with variance and skewness (mean, or lower bound of the support) of underlying WTDs. Detailed numerical results can be found in Supplementary Figs 24–30.

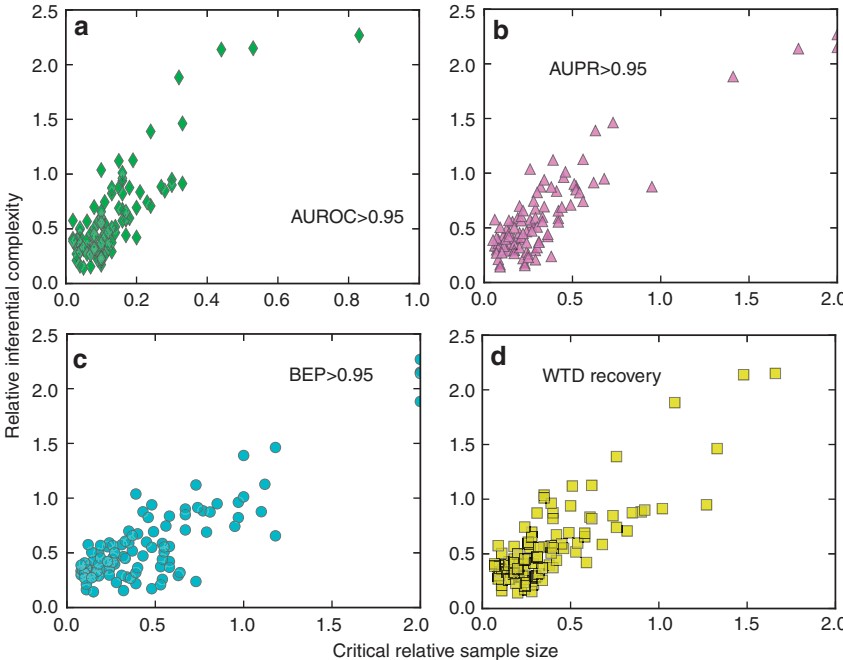

**Figure 4 | Inferential complexity of benchmark STNs. (a–d)** Scatter plot of relative inferential complexity [$\equiv N^{-1}\langle\xi(\mathcal{D},\mathcal{N}_s)\rangle_D$] versus critical relative sample size for ensuring at least 0.95 AUROC, AUPR and BEP, respectively, as well as that for simultaneously recovering both the underlying networks and WTDs. Here, the angular brackets $\langle\cdot\rangle_D$ denote the average over ensembles of diffusion cascades. Each point represents an average over at least ten independent realizations with respect to a specific combination from benchmark networks and WTDs, and the critical values of respective relative sample sizes are listed in Supplementary Table 5. Numerical results show that there is positive correlation between the inferential complexity and the critical relative sample size for achieving network reconstruction at a given accuracy, implying that the more complex (or indeterminate) the underlying diffusion structure, the more difficult the inference of temporal networks from data.

## Discussion

In summary, we have developed a general framework to reconstruct STNs as a null model of temporal networks by fitting time-course observations of the diffusion process taking place on them. To alleviate the ill-posedness of this time-extended inverse problem, we have decomposed the task of network reconstruction into structural and temporal aspects, that is, an unsupervised topology inference for the time-aggregated networks using MCMC sampling, and a nonparametric density estimation for the associated WTDs via self-consistent iterations, respectively. We have given a rigorous consistency proof for the proposed WTD estimator, and have numerically shown that the iterative algorithm frequently possesses good convergence properties given properly selected, data-driven initial guesses. We have also applied our method to various types of benchmark STNs and empirical temporal networks, showing that it is statistically possible to discover latent temporal networks with high accuracy only from a moderate amount of diffusion data. Despite the inability to recover actual serial snapshots of networks, the reconstructed STN builds up both structural and temporal statistics sufficient to enable reliable prediction of the population-level diffusion behaviour, which can thus be of practical interest in a broad range of applications, particularly in time-critical and privacy-sensitive circumstances related to human activities.

Our reconstruction method is not only restricted to the data generative process considered in the paper, but is also adapted to many variants of the original inference task. For example, recent work[27] has demonstrated the predictive power of the Hawkes process method for quantifying and tracing information cascades on social media, provided a well-defined memory kernel (a mathematical equivalent to the WTD in our model) fitted to the empirical mechanism for diffusion. Our method then provides an alternative operative route for estimating the memory kernel from early time courses of diffusion, as well as for updating the posterior WTD self-consistently with new available process data as diffusion proceeds, while not necessitating access to possibly privacy-sensitive information at the individual level. Another important extension is to use observed time courses of disease outbreaks to deduce the underlying population structure and temporal transmission pattern in epidemiological studies. Here, the essential difference between a generic epidemic process and the information broadcasting considered here lies in that the recovery process (for example, self-healing) prevents to a large extent the possible transmission of diseases from infected individuals to the remainder of the population. Restated in the language of survival analysis, the observation of successful transmissions is potentially 'censored' by the recovery events occurred among the infected population. Under this circumstance, one can directly make use of subdistributed WTDs (that is, $\int_0^\infty \rho_{uv}(\tau)\mathrm{d}\tau < 1$) to encapsulate arbitrary distributions of transmission and recovery times[30], and apply the same inferential scheme with few modifications. We note here that the residual probability mass located at the infinity, $r_{uv} = 1 - \int_0^\infty \rho_{uv}(\tau)\mathrm{d}\tau$, meaning the probability of endemic transmissions being interrupted by node $u$'s recovery, causes a decrease in the effective number of diffusion cascades that are helpful for predicting if link $(u,v)$ is present—as a consequence, the larger $r_{uv}$, the less observably succeeded transmission routes, the more diffusion data required to extract the latent network. To exclude this nuisance factor we therefore have focused on the information diffusion model that mimics susceptible-infected epidemics (with $r_{uv} = 0$) to capture the effective sample sizes for specific reconstruction goals.

In addition to the simplified forward diffusion model, we have made several assumptions in this paper, of which the most

important is that the STNs are assumed to have no cross-dyad interdependence. This is the key property that enables the localized computation of likelihoods and branching coefficients using only neighbourhood information, thus ensuring an efficient implementation with polynomial complexity. We have discussed the second-order STN model that assumes a joint distribution of waiting times occurring on any successive dyads. Such polyadic correlations destroy the applicability of Matrix-Tree Theorems, leading to the necessity of a tree-enumeration procedure with exponential complexity. Leaving aside temporarily the issue of implementation expenditure, our inferential scheme can be naturally extended to the higher-order correlated case (see Supplementary Note 9). We have furthermore devised a pairwise approximation of branching coefficients in order to speed up our algorithms. Another significant one, mainly adopted in our benchmark tests, is the homogeneous population assumption, stating that the underlying WTDs $\rho_{uv}(\tau)$ are identical, despite the individual heterogeneity or frailty widely prevalent in many real-life populations. Again, it should be pointed out that although our framework is applicable to non-identical WTDs, as well as higher-order, polyadically correlated cases, the established STN model to some extent sacrifices simplicity and runs the risk of overfitting. The ultimate goal of null modelling is to reproduce expected patterns that capture predictive information in data, while not being overly complicated.

Let us further inspect the complexity and scalability of our method. While it is generally difficult to quantify the complexity of the iterative estimator even for the first-order STNs, benchmark test results numerically illustrate that the proposed procedure, when initialized with appropriate first guesses (see Supplementary Note 6), has quite good iterative convergence, irrespective of either structural properties of the time-aggregated topologies or distributional shapes of the dyad-specific WTDs. Thus our algorithm can practically be implemented with complexity $O(CN^2M)$, where $C$, $N$ and $M$ are the numbers of diffusion cascades, network nodes and MCMC iterations, respectively. Because $M$ only exerts influence on the precision of a posteriori prediction of links and is independent of the problem size, the actual complexity of our procedure is then $O(CN^2)$, which reaches the theoretically optimal time complexity. Put another way, the network reconstruction problem can be regarded as a sequence of $O(N^2)$ likelihood-ratio hypothesis tests, in which the marginal gain of each possible link calculated using $O(C)$ diffusion cascades is utilized to decide if the corresponding dyad of nodes is truly connected. Despite the proposed method with the (near-)inherent, quadratic complexity, there is a widespread need for more scalable and efficient, hopefully linear-time, inverse modelling approaches for empirical complex systems, especially in the era of Big Data. The potential solution to this impasse is to shift from predicting individual links to zooming out to the coarse-grained or mesoscopic topological scales using more sophisticated inference techniques borrowed from, for example, large-scale hypothesis testing[43] and community-based time series analysis[44].

There are also numerous conceivable approaches for our topology inference step. A closely related alternative one, which is formulated in a Bayesian fashion, is to introduce prior information about the underlying diffusion network into the estimation scheme to encompass a broader spectrum of expected structural features. Here we have exemplified this with the standard $\ell_1$-norm sparsity priors, turning the maximum likelihood into a maximum a posteriori (MAP) estimation by virtue of slight differences in calculating the marginal gains of network links (see Supplementary Note 12). Numerical results have demonstrated the significantly improved reconstruction performance for STNs. Furthermore, parametric network models (such as exponential random graphs[45]) and advanced optimization

techniques for the MAP problem, as well as practically available prior information other than collective diffusion data (for example, partially observed topology structure, relevant metadata of network agents, and so on) can also be incorporated into our reconstruction scheme with little technical effort.

Another contribution of the proposed inferential framework is the density estimation for underlying WTDs. Because of the context in which we are interested, an ideal null model of temporal networks would be reproductive of time variability in empirical interactions, as well as predictive of their collective dynamical properties at the desired level, in particular with the help of the WTD extracted from true latent (usually anomalous) diffusion patterns. When the goal is to solely create temporality information with prior network structures, rather than to reconstruct the topology of diffusion substrates, the density estimation can be pursued by a radically sharpened diffusion cascades. Under the homogeneous population assumption, meaning a population-wide fitting of the WTD commonly shared by all underlying dyadic interactions, one can expect a reliable WTD estimation even from a single diffusion cascade, unless either network size or observation window is very small. Consequently, our method has potential applications in epidemiological studies, especially for inverse modelling of general epidemics with non-Poissonian behaviour[30], which range from estimating parameters such as the effective reproductive number using subdistributed WTDs to identifying the invasion routes of epidemics using branching coefficients. Finally, the central role played by branching coefficients—in the universal imputation scheme for treating the censored data problem arising from competing risk events of networked diffusion—has implications for definition and refinement of the dynamics-based node or link centrality and information-theoretic complexity measures for temporal networks.

## Methods

**Benchmark test data.** To test the performance of our inference algorithm, we have carried out extensive numerical experiments using three categories of time-aggregated benchmark topologies (deterministic, stochastic and empirical static networks) in combination with several different types of benchmark WTDs, as listed in Supplementary Tables 1 and 3. To provide further empirical validation, we also apply our method to several realistic temporal contact networks, as listed in Supplementary Table 2. Specifically, we reconstruct the STN model from synthesized diffusion cascades respectively using empirical contact lists and the STN tuple that is fitted to the original data (see Supplementary Table 4 for detailed experimental settings). Network data or their generative models can be found in relevant references.

**Gibbs sampler for time-aggregated graphs.** We apply MCMC method with Gibbs sampling to explore the configuration space of the underlying time-aggregated network. The configuration transition is based on the link-flipping operation with success probability according to the likelihood ratio between the new configuration and the old configuration, that is, the marginal gain of the flipping operation (equation (6)). In our benchmark test, the burn-in period and the maximum lag are set to 10, and the number $M = 200$ of MCMC samples are drawn. See Supplementary Note 7 for detailed algorithm implementation.

**Iterative procedure for the WTD estimator.** To solve the self-consistent equations for $\hat{\rho}(D, \mathcal{G})$, the algorithm carries out an iterative procedure as follows: Start with a priori guess for the underlying WTDs $\hat{\rho}^0 = [\hat{\rho}^0_{uv}]_{(u,v) \in \mathcal{G}}$ and repeat the following synchronous updates for all entries of $\hat{\rho}^k$ until convergence

$$\hat{\rho}^{k+1}_{uv}(\tau) = \frac{1}{C} \sum_{i=1}^{C} \left\{ \begin{array}{l} \beta_{uv}(\mathcal{D}^i, \mathcal{G}, \hat{\rho}^k) K_h(\tau - d^i_{uv}) \\ + \left[ 1 - \beta_{uv}(\mathcal{D}^i, \mathcal{G}, \hat{\rho}^k) \right] \frac{\hat{\rho}^k_{uv}(\tau) \Theta(\tau - d^i_{uv})}{\hat{\Phi}^k_{uv}(d^i_{uv})} \otimes K_h(\tau) \end{array} \right\}. \quad (9)$$

The convergence is judged by a small change in the estimated $\hat{\rho}^k$ between successive iterations. Here we adopt the Kolmogorov–Smirnov divergence, and terminate the iteration whenever $\| \hat{\rho}^{k+1}, \hat{\rho}^k \|_{KS} = \max_{(u,v) \in \mathcal{G}} \max_{\tau} | \hat{\Phi}^{k+1}_{uv}(\tau) - \hat{\Phi}^k_{uv}(\tau) | < \varepsilon$, where $\hat{\Phi}^k_{uv}(\tau) = \int_{\tau}^{\infty} \hat{\rho}^k_{uv}(t) dt$ is survival function of WTD $\hat{\rho}^k_{uv}(\tau)$, and $\varepsilon$ is a predetermined error threshold. The detailed self-consistent iterative scheme is presented in Supplementary Note 7.

In our numerical validation, we assume for brevity that internal temporal interactions among a population satisfy the homogeneity condition, corresponding to first-order the STN with i.i.d. dyad-level WTDs. To initialize the iterative procedure, we use exponential WTD as the first guess, and set smoothing kernel bandwidth $h = 0.05$. Practical parameter selection criteria, as well as their effects on convergence properties and performance of the density estimator are discussed in Supplementary Note 6.

**Notations.** We summarize the notations used throughout this paper in Supplementary Table 6.

**Data availability.** All relevant data are available from the authors on request.

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

## Acknowledgements

We thank Yi-Qing Zhang and Cong Li for fruitful conversations. This work was supported by the National Natural Science Fund for Distinguished Young Scholar of China (No. 61425019), the National Natural Science Foundation (No. 61273223), the National Natural Science Foundation of Shanghai (No. 16ZR1446400) and Shanghai SMEC-EDF Shuguang Project (No. 14SG03).

## Author contributions

X.L. and X.L. designed and performed research. X.L. analysed the data and wrote the paper. All the authors edited the manuscript with critical review.

## Additional information

**Competing interests:** The authors declare no competing financial interests.

