## [Peer Review File · Nature Communications]

Reviewer #1 (Remarks to the Author):

This paper studied the problem of reconstructing temporal contact network features by using only the information of the first arrival times of diffusion cascades at network nodes. The assumption is that contacts between nodes follow a stochastic temporal network (STN) model, in which links between nodes comprise a static network and contacts occurring on a link are generated by a renewal process with a given interevent time distribution. Therefore, the reconstruction problem is mapped to the maximum likelihood inference of the STN model given the first arrival times of cascades. The performance of the proposed algorithm is evaluated with several performance measures, for the data generated by the benchmark STN models with various network structure and the interevent time distributions.

The mathematical techniques employed in this paper are solid and theoretically sound. The language and the presentation of the results meet the standard criteria for publications in top journals. My main concern is about the problem setting itself and its attractiveness to general audience (see Major point below). Mainly because of this issue, I do not recommend the present manuscript for a publication in Nature Communications. However, I hope that the authors take into account the comments described in the following to improve the scientific impacts of the manuscript.

[Major point]

1. Problem setting

In this paper, the authors assumed that the contact processes between nodes underlying diffusion cascades are generated by the STN model. I agree that the STN model has an advantage of its simplicity on theoretically investigating diffusion phenomena on temporal networks. However, the aim of this study, that is the reconstruction of temporal networks, is more practical than purely theoretical problems. Thus, I wonder how much of interest it is to infer the STN models.

In the Introduction, it is stated that reconstructing temporal networks as the serial snapshots can only be possible under strong and arguably unrealistic assumptions. I agree with this point, however, I also think that the same criticism holds true for the STN model too. The STN model stands on very strong assumptions if we regard this as a model of empirical temporal networks. The model assumes that the contact sequences on links are temporally and structurally independent of each other. The temporal independence means that the IETs on a link are generated in the i.i.d. manner and thus the contact sequence on a link is an instance of the renewal process. The structural independence means that the contact sequences of any two links are independent of each other. However,

research communities have noticed that such notions of independence do not hold for empirical temporal networks and that the correlations in contact sequences are important factors when we consider dynamical processes on temporal networks (e.g., Karsai et al., *Scientific Reports* 2, 397 (2012); Vestergaard et al., *Physical Review E* 90, 042805 (2014)). Therefore, I do not think that the inference of the STN model overcomes the issue about the inference of previous methods based on serial snapshots and that the inference of the STN model provides valuable insights about the temporal features of empirical temporal networks. To claim the practical performance of the proposed framework for empirical settings, I recommend carrying out the numerical simulations on empirical temporal networks. To be more precise, it is to apply the proposed method to the first arrival times of cascades simulated on empirical contact sequence data and to interpret the derived results. The empirical "static" networks are considered as the substrate of the STN model as shown in Table 1, but this tells nothing about the effects of correlations that we generally observe in empirical "temporal" networks. I believe that this additional examination improves the scientific impact of this study.

I have another concern about the problem setting, especially on the privacy protections. In several parts of the manuscript, it is stated that the proposed method can achieve complete privacy protection for individuals that methods based on time-resolved interaction data cannot. In other words, it is supposed that time-resolved interaction data (i.e., who interacts whom at when) are private and inaccessible while the first arrival time data (i.e., who adopts something at when) are public and accessible. However, I cannot imagine any real situations that satisfies this assumption, and I do not agree that the first arrival time of cascades are not private. Should be discussed some real examples of such situations that support this assumption and the justification of this distinction between private and public information.

[Minor points]

1. Writing

The precise problem setting was not clear to me until I read through the manuscript again for several times. One reason of this hard readability comes from that the formal definition of the problem -- what information we are given and what we want to estimate about the system-- was stated for the first time in p.8. Therefore, structure of the manuscript should be improved to provide readers with clearer introduction to the problem considered.

2. Terminology

Some unusual technical phrases are used without definitions, such as "time-extended internal organisation" (line 20) and "the static closure graph structure" (line 103). Proper definitions should be provided for better readability.

[Minor, technical questions]

1. Lines 123-124. It is stated that the branching coefficients serve a "soft" rather than sharp censoring indicator. I do not understand what it implies and further explanation is needed.

2. Line 152. The symbol "N" is used to represent a temporal network, however, I do not agree with this idea because "N" conventionally represents the number of nodes (the size of V in the terminology of this paper). It should be considered to replace it with other suitable symbols to avoid unnecessary confusion of readers.

3. Line 157. Strictly speaking, the temporal event (u, v, t) is not in N but in E, and topological link (u, v) is not in G but in the set of edges of G. Such mathematical expressions should be carefully revised.

4. Line 170. I do not get what the subscript "n" stands for. If "n" represents the number of events comprising the time-respecting path, then the set considered should not be E but E^n .

5. Lines 190-191. Is T actually tree or a directed acyclic graph with some loop if we discard the link directions?

If the latter holds true, calling it a tree is confusing.

6. Lines 203-204. I do not understand why L^n is a triangular matrix up to similarity permutation. Is G supposed to be a directed acyclic graph?

7. Line 230-231. It is stated that the Gibbs sampler iteratively flips any links. Is this mean to switch the presence and absence of a link in G?

8. Line 265. The waiting-time distributions considered are shown in Fig. 3. I am concerning that all the distributions have a narrow support (effectively) in $0 < \tau < 4$. In reality, the waiting time distributions usually possess long tails which has considerable impacts on dynamical processes on temporal networks. Therefore, I want to see how the proposed method works when the waiting-time distribution follows a skewed distributions.

9. Lines 269-270. The figure S6 indeed looks similar to a first-order phase transition, but further statistical-physics analysis is needed if it is said with absolute certainty. Otherwise the expression should be removed or weakened.

10. Lines 276-277. I do not get why the adoptive mechanism for the fitting explains the smaller number of cascades required for inference than expected. Further explanation is needed.

11. Line 325-326. It is stated that the reconstructed STN gives statistics sufficient for reliable prediction, which I do not agree. It is shown in the manuscript that the reconstruction methods performs well for the STN benchmarks, however, no prediction is made through this procedure. Thus, additional evaluation should be provided to demonstrate the predictive power of the proposed method.

[Comments on figures and tables]

Fig 1. The panels (a)-(d) are difficult to see as they are distorted. Alternative way of their presentation is recommended.

Fig 2. In caption, the notions of "incorrectly existing links" and "missing links" are not defined and hard to understand. I suggest to replace them with more common notions like "false-positive links" and "false-negative links".

Fig 4. It seems that the panels (a)-(c) are not mentioned in the main text. If so, these panels should be removed. If they should be kept, at least the legends of them should be enlarged to be readable.

Table 1. I do not get the idea of the right-bottom element of each cell, "simultaneous reconstruction of both underlying networks and associated WTDs". Is that what was done throughout this study, isn't it? In addition, relative sample size is mentioned in the caption but not presented in the table.

Table 2. Should be provided the exact correlation coefficient values for these factors, beyond their tendency of positive or negative correlations.

Reviewer #2 (Remarks to the Author):

Dear Editor,

In this paper, the authors focus on the important problem of reconstructing structural and temporal patterns from empirical observations in networked systems. To do so, the authors adopt the perspective of renewal processes, or stochastic temporal networks, allowing to take into account the heterogeneous inter-event times observed in empirical systems, while imposing some statistical constraints. This work brings without hesitation an important contribution to the active field of temporal networks. The authors mostly focus on situations where adjacent edges are independent, but also describe the situation where correlations are non-negligible. I am overall very positive about this work, clearly written, and whose theoretical contribution is sound and consequent. However, I am less convinced by the empirical validation, mainly performed on artificial data. If possible, I would advise the authors to test the method on real-world temporal networks, where some of the modelling assumptions are not verified (e.g. non-stationarity). The authors should also draw connections with different works adopting a different, yet related approach for network reconstruction and cascade size predictions, using Hawkes processes (very similar to random walks on stochastic temporal networks). Examples include:

<http://snap.stanford.edu/seismic/seismic.pdf>

<https://www.mpi-sws.org/~manuelgr/pubs/S2050124214000034a.pdf>

After these minor points are addressed, I will be very happy to recommend the publication of this work in Nature Communications.

Reviewer #3 (Remarks to the Author):

The paper studies the problem of reverse-engineering the structure of temporal networks when detailed temporal data on dyadic events is missing. It develops an efficient algorithm for doing so, using a stochastic null model of diffusive arrival times, assuming no dependence between different dyadic interactions. Benchmark tests applied to synthetically generated networks and to toy time-aggregated networks show that temporal networks can be unveiled with high accuracy.

I think the paper is well researched: it cites all the recent temporal network literature and its applications of which I am aware adequately, and makes good points why the problem of

reconstructing temporal networks is important, such as in epidemiology. Clearly there are broad applications to the problem. Concerning language, the paper is also well written. It is good that the case of correlations of events between different dyads is discussed, as this would have been my first objection in terms of limitations. As having not the most technical expertise in this specific sub-field, I was not able to follow all the technical derivations, but from what I can tell they seem correct.

The title and promise of the paper is very interesting, however I had 2 major problems with the manuscript. These might be problems that can be fixed, but it is not clear to me as the lack of overall clarity is one part of the problem.

One issue concerns only the presentation and outline of the big picture. A problem for me was the low readability of the Results section due to the extreme technicality of the subject. In other words, the "Why are we doing this?" and the "So what?" questions often remain unanswered, which I believe should not be the case in a non-specialist journal. For example, I can see the different steps that build up the algorithm, but why are they important? The sub-sections read "We next calculate.., We now introduce.., This step takes.., This step takes.." without saying "This is important because..". I understand that most of the mathematical derivations have been deferred to the SI, which is good, but still, I feel the manuscript's main results remain intractable to the general reader. Certainly I would not expect a general reader to understand everything in the technical development of such a complex algorithm, but somebody from the network science community should at least be able to get half of it. I doubt this is the case in the current presentation. I am sorry my criticism is so vague, but the vagueness comes from not fully understanding from the manuscript the importance of what the steps of the model do to advance previous approaches or the field itself. For example, maybe the point on privacy could have been elaborated better.

Minor issues on clarity are also present in the figures. For example, the toy network used in Figure 1 is not so simple, and the concepts it aims to explain are lost because of that. In this figure, the ROC curves are shown, but the text or the caption do not deliver the message of what this actually means or why it is important. It is not clear what the legend refers to (#1, #2,..).

The second issue I see is the lack of tests with relevant (large) data sets from real systems. In a specialized theory-oriented journal, the vast array of benchmarks and small, time-aggregated toy networks which were tested, would be more than enough. However, in an important general paper on reconstructing temporal network structure, I would expect the test of the proposed model on a real data set where detailed link events are available, and comparison of the reconstruction results to the case where the time information was aggregated. This data could be mobile phone calls, internet messages/emails, or any of the large-scale temporal data sets available online, for example from <http://www.sociopatterns.org/datasets/> or from <http://projects.csail.mit.edu/dnd/>. These are just first results from a quick internet search, I am sure there are more data sets available publicly.

In summary, the authors have clearly put a lot of effort into the mathematics and seem to have done a great job with the technical details, but the main results and their consequences are too much hidden behind the technical nature of the presentation which makes it hard to judge the paper's impact.

Response for NCOMMS-16-18608

Dear Editor and Reviewers,

We gratefully appreciate your comments and helpful suggestions, based on which we have revised our manuscript. Following the advice of the editor and reviewers, we have provided empirical validation of our method using additional datasets from real-world temporal networks. Also, in order to enhance the manuscript's readability, we have made our best attempts to motivate and illustrate the methodology we developed in this study, with special care taken to present the algorithm structure as well as the obtained results in a more approachable manner. Our point-by-point responses to the editor and the reviewers can be found in the attachment. We hope that we have adequately answered the reviewers' questions and the revised manuscript is satisfactory now.

Thank you for your help with processing our paper.

Best wishes,

The Authors: Xun Li, Xiang Li, Cong Li

Responses to the First Reviewer

Comment - This paper studied the problem of reconstructing temporal contact network features by using only the information of the first arrival times of diffusion cascades at network nodes. The assumption is that contacts between nodes follow a stochastic temporal network (STN) model, in which links between nodes comprise a static network and contacts occurring on a link are generated by a renewal process with a given interevent time distribution. Therefore, the reconstruction problem is mapped to the maximum likelihood inference of the STN model given the first arrival times of cascades. The performance of the proposed algorithm is evaluated with several performance measures, for the data generated by the benchmark STN models with various network structure and the interevent time distributions.

The mathematical techniques employed in this paper are solid and theoretically sound. The language and the presentation of the results meet the standard criteria for publications in top journals. My main concern is about the problem setting itself and its attractiveness to general audience (see Major point below). Mainly because of this issue, I do not recommend the present manuscript for a publication in Nature Communications. However, I hope that the authors take into account the comments described in the following to improve the scientific impacts of the manuscript.

Response - First of all, we are grateful for your positive comments to our manuscript. We understand this concern about “general interests” of non-specialised readership, and during our revision of this paper, considerable attention has been therefore paid to the attractiveness and readability of our paper, in particular for the introductory part. We have answered your questions point by point (see below) and have revised our manuscript in line with your suggestions.

Comment - [Major point]

1. Problem setting

In this paper, the authors assumed that the contact processes between nodes underlying diffusion cascades are generated by the STN model. I agree that the STN model has an advantage of its simplicity on theoretically investigating diffusion phenomena on temporal networks. However, the aim of this study, that is the reconstruction of temporal networks, is more practical than purely theoretical problems. Thus, I wonder how much of interest it is to infer the STN models.

Response - Thank you for this comment. As you pointed out, the ultimate but also idealised goal is to create perfect recovery of latent temporal networks from data, and the STN model is indeed, in many practical occasions, an obvious oversimplification unless realistic networked systems could have been composed of identical, independently interacting agents. Nevertheless, we claim that it is of interest to infer the STN models from a statistical perspective primarily due to the following two reasons.

First, reconstructing the underlying time-stamped interaction sequences (or, equivalently, serial snapshots of temporal networks) is mathematically ill-posed, the solution of which is usually very difficult, if not absolutely impossible. Instead, with the help of the assumptions of STN models, the inferential results—the estimated first-order WTD $\hat{\rho}(\tau)$ commonly shared by the STN links (also see next response for more justification of this point, page 3)—can be thought of as the mean-field distribution of waiting-times averaged over the entire network’s interactions. Put another way, inference of the STN model offers us the choice of accuracy level according to the sample size—when the available data are too few to afford perfect reconstruction, one can then choose to estimate the mean-field WTD

$\hat{\rho}(\tau)$. Also, with sufficient data at hand one can optionally switch to a finer level to explore, for example, correlation between waiting-times of adjacent links by employing the higher-order STN model with joint conditional WTD $\hat{\rho}(\tau_1|\tau_2)$. In the thermodynamic limit of large sample size, it follows from the proof of the estimator $\hat{\rho}$'s asymptotic consistency that our method can faithfully recover each pairwise WTD, $\rho = [\rho_{uv}(\tau)]_{(u,v) \in G}$, thus corresponding to the most generalised case in which network agents have heterogeneous characteristics. Therefore, the proposed framework based on STN models is competent for statistical inference of realistic temporal networks at different levels of approximation in accordance with the given sample size (also see our responses on page 12 for further empirical evidence). This could thus be of particular interest in the practically relevant regime of sparse data.

Second, the STN model provides a good null modeling of empirical temporal networks to capture the regularity of latent interaction patterns from both structural and temporal aspects. In such a context, estimating the time-aggregated graph G and the population-level WTD ρ of a temporal network is even more important than detailing the underlying individual interactions (u, v, t) . It is like a physician who only gauges the temperature of liquid water, thus avoiding the unnecessary derivation of real molecular motions from Newtonian mechanics (which is also infeasible). Similarly, thanks to their simplicity and mean-field nature, the STN models are not only of theoretical interest, but of statistical predictive power in many practical situations (also see page 9 for our relevant response to the issue on “predictive power”). Recently years have witnessed increasing effort to understanding real-life diffusion phenomena on temporal networks. Despite being one of the most fruitful idea, these STN-based analytical methods seem, at least at first sight, to be purely theoretical and incompatible with the mainstream of empirical research on temporal networks. It is presumably due to the so-called “model selection” issue, that is, there is lack of appropriate methodologies for assessing whether and which “parameters” of the STN model $N = (G, \rho)$ ¹ fits empirical data. Here we aimed to take a step towards filling this gap: through statistical inference techniques we reconstruct STN models for empirical temporal networks so that the many analytical tools can be applied to the inferred STNs to predict the collective diffusion behaviour of the networked systems. In summary, we believe inferring STN models is of general interest for both theoretical and practical investigations on temporal networks.

Comment - In the Introduction, it is stated that reconstructing temporal networks as the serial snapshots can only be possible under strong and arguably unrealistic assumptions. I agree with this point, however, I also think that the same criticism holds true for the STN model too. The STN model stands on very strong assumptions if we regard this as a model of empirical temporal networks. The model assumes that the contact sequences on links are temporally and structurally independent of each other. The temporal independence means that the IETs on a link are generated in the i.i.d. manner and thus the contact sequence on a link is an instance of the renewal process. The structural independence means that the contact sequences of any two links are independent of each other. However, research communities have noticed that such notions of independence do not hold for empirical temporal networks and that the correlations in contact sequences are important factors when we consider dynamical processes on temporal networks (e.g., Karsai et al., Scientific Reports 2, 397 (2012); Vestergaard et al., Physical Review E 90, 042805 (2014)). Therefore, I do not think that the inference of the STN model overcomes the issue about the inference of previous methods based on serial snapshots and that the inference of the STN model provides valuable insights about the temporal features of empirical temporal networks.

¹Although in our model settings, G is assumed to be non-specific network configuration (in contrast to, for example, the family of exponential random graphs) and ρ to be non-parametric WTD (in contrast to parametric distributions), they can still be thought of as being selected from some generalised parameter spaces of the STN model.

Response - Yes, we have also noticed several important works investigating on how the correlation in contact sequences affects dynamical processes on temporal networks. We are also of the same view that imposing structural independence often leads to markedly deviations when predicting dynamical behaviour of networked systems. It is indeed an important issue to further justify the model assumptions used in our manuscript. On one hand, as we discussed in our response to your previous point, the proposed inferential scheme is also applicable to higher-order STN models which encapsulate such inter-link correlations, whenever the collected data are sufficient for inference at the corresponding levels. On the other hand, the key assumption of identically distributed IETs is originated from Ockham's principle of parsimony—despite the gap between realistic systems and their null models, a statistician is always inclined to select the simplest model with the minimum number of parameters to be inferred from data, unless particular reasons are invoked for not doing so. In our situation, it is reasonable to adopt the homogeneous population assumption, rather than being non-identically distributed WTDs, to explain the observed diffusion cascades, because no clue to the heterogeneity of network agents is offered by raw data. Therefore, our model assumption is natural and common in many statistic inference applications, especially when the sampled data size is relatively small. Besides, we have numerically synthesised diffusion cascades on several empirical temporal networks in order to examine the these ignored effects in the validity and performance of our inferential method (also see our next response on page 12).

As a final remark, using the STN model permits, at least in the asymptotic limit, perfect reconstruction of heterogeneous WTDs, while it is worth stressing on the overfitting problem leading to overly complicated models.

Comment - To claim the practical performance of the proposed framework for empirical settings, I recommend carrying out the numerical simulations on empirical temporal networks. To be more precise, it is to apply the proposed method to the first arrival times of cascades simulated on empirical contact sequence data and to interpret the derived results. The empirical “static” networks are considered as the substrate of the STN model as shown in Table 1, but this tells nothing about the effects of correlations that we generally observe in empirical “temporal” networks. I believe that this additional examination improves the scientific impact of this study.

Response - Thank you very much for helpful suggestion. Following your advice, we have used additional empirical datasets from real-life temporal networks to support our methodology. Since similar comment was shared by other reviewers, please refer to, for example, pages 12 and 15 for our relevant responses.

Comment - I have another concern about the problem setting, especially on the privacy protections. In several parts of the manuscript, it is stated that the proposed method can achieve complete privacy protection for individuals that methods based on time-resolved interaction data cannot. In other words, it is supposed that time-resolved interaction data (i.e., who interacts whom at when) are private and inaccessible while the first arrival time data (i.e., who adopts something at when) are public and accessible. However, I cannot imagine any real situations that satisfies this assumption, and I do not agree that the first arrival time of cascades are not private. Should be discussed some real examples of such situations that support this assumption and the justification of this distinction between private and public information.

Table 1: **Summary of privacy-sensitivity related synonyms**

Algorithm I/O	Sensitivity	Content specification	Level	Direct access	Ad hoc problem setting
Input	Public	Collective dynamics	Path-level	Yes	Diffusion cascades \mathbf{D}
Output	Private	Individual interaction	Link-level	No	STN model $N_s = (G, \rho)$

Response - We first motivate our problem setting by the following two examples. In viral marketing, one important task is to access the so-called “network value” of a customer. It measures the extent to which the customer, in addition to buying products himself, may influence others to buy them via, for example, word-of-mouth recommendations. Therefore mining social networks is of critical value to commercial companies for making optimal marketing decisions [1]. In such situations, word-of-mouth communications (i.e., when and by whom customers are persuaded to buy products) can be thought of as privacy-sensitive interactions among customers, while the first-arrival-time data (i.e., when customers purchase products) are readily available to companies, while containing no private information of customers. The second example is information diffusion in blogspace [2]. When tracking an piece of information propagating across a social network, the first-arrival-time data are obviously public because the weblogs containing the information of interest are time-stamped, and in the contrary, the weblog owners may occasionally hide the information source for private concerns, making implicit their underlying social interactions that spreads the information of interest. We have elaborated this privacy-protection issue by such realistic examples in the revised manuscript.

Also, there are many other conceivable examples such as infectious diseases² (see, e.g., our previous work [3]), contagious memes, Facebook posts and tweets (see, e.g., Refs. [4, 5] provided by the second reviewer). We are now able to justify the distinction between private and public information. In short, it depends upon the subjective willingness of network agents to provide this information, as well as the objective accessibility of the data to a statistician for fulfilling the inference task. Especially, despite the individual interaction data being privacy-sensitive to gather, the many publicly observable data can be collected from a variety of dynamical processes associated with collective behaviour of the networked systems, as summarised in Tab. 1.

Remark.—Finally, although seemingly irrelevant to our problem setting, we wish to point out an entirely similar situation that arises from the engineering literature of network tomography. In diagnostics of communication networks (such as Internet [6]), many individual link-level parameters, concerning package loss and delay³ as well as topology structure of the underlying network, have been proved technically identifiable from end-to-end, path-level measurements (which are equivalent to the data of time difference of arrivals, TDOA). Interestingly, much research effort has also been devoted to the so-called “active tomography” techniques, aiming at the optimal design of, for example, flexicast probing experiments (i.e., determining the location of the various sources where end-to-end losses and/or delays are measured on the network [7]) for the inferential purpose. At the end of our response to this privacy issue, we believe that the ideas from active tomography offer a deeper un-

²More specifically, the daily surveillance report of the epidemic (i.e., patients’ hospitalization dates) can be thought of as (partially) observed diffusion data, from which the underlying contact network that spreads the disease (or sometimes the epidemic pathways embedded therein) are to be inferred.

³However, the majority of related works assumes a very simple form of link-level parameters for package loss and/or delay, e.g., single link loss rates, exponential or Gamma delay distributions, to name a few. Our work has extended this to the most general case in a nonparametric setting.

derstanding of privacy sensitivity, and instead of passive inference using given public data, elaborate design strategies (wisdom of crowds⁴) of diffusion cascades toward inference for temporal networks deserve future study.

- [1] Domingos, P. (2005). Mining social networks for viral marketing. *IEEE Intelligent Systems*, 20(1), 80–82.
- [2] Adar, E., & Adamic, L. A. (2005). Tracking information epidemics in blogspace. *Proceedings of IEEE/WIC/ACM International Conference on Web Intelligence*, 207–214.
- [3] Li, X., Li, X., & Jin, Y. Y. (2012). A Data-driven inference algorithm for epidemic pathways using surveillance reports in 2009 outbreak of influenza A (H1N1). *Proceedings of IEEE International Conference on Decision and Control*, 2840–2845.
- [4] Rodriguez, M. G., Leskovec, J., Balduzzi, D., & Schölkopf, B. (2014). Uncovering the structure and temporal dynamics of information propagation. *Network Science*, 2(01), 26–65.
- [5] Zhao, Q., Erdogdu, M. A., He, H. Y., Rajaraman, A., & Leskovec, J. (2015). Seismic: A self-exciting point process model for predicting tweet popularity. *Proceedings of ACM International Conference on Knowledge Discovery and Data Mining*, 1513–1522.
- [6] Coates, A., Hero III, A. O., Nowak, R., & Yu, B. (2002). Internet tomography. *IEEE Signal processing magazine*, 19(3), 47–65.
- [7] Lawrence, E., Michailidis, G., & Nair, V. N. (2006). Network delay tomography using flexicast experiments. *Journal of the Royal Statistical Society: Series B*, 68(5), 785–813.
- [8] Daneshmand, H., Gomez-Rodriguez, M., Song, L., & Schölkopf, B. (2014). Estimating diffusion network structures: Recovery conditions, sample complexity & soft-thresholding algorithm. *Proceedings of IEEE International Conference on Machine Learning*, 793–801.

Comment - [Minor points]

1. Writing

The precise problem setting was not clear to me until I read through the manuscript again for several times. One reason of this hard readability comes from that the formal definition of the problem—what information we are given and what we want to estimate about the system—was stated for the first time in p.8. Therefore, structure of the manuscript should be improved to provide readers with clearer introduction to the problem considered.

Response - Thank you for advice. We have taken special care to the statement of our problem. Using a more illustrative style, we have enhanced the readability of the introductory part of our manuscript (see, e.g., the updated Figure 1).

Comment - 2. Terminology

Some unusual technical phrases are used without definitions, such as “time-extended internal organisation” (line 20) and “the static closure graph structure” (line 103). Proper definitions should be provided for better readability.

Response - Thank you for the reminder. We have replaced these inaccurate (or at least inadequately interpreted) phrases with more common ones to enhance our manuscript’s readability. More precisely,

⁴Finding approximate algorithms, in particular with fast implementation, solving this problem is extremely difficult, let alone the optimal solution. (See, e.g., Ref. [8] for theoretical discussion on the recovery conditions, as well as the lower bound of sample size allowed for inferring temporal networks.) Fortunately, there are numerous conceivable heuristics, e.g., to cover the broader area of the underlying network, it is better to select distantly apart than adjacent nodes as sources of the diffusion cascades. Such choice targets to generating a set of minimally overlapped diffusion cascades (at least at their early stages) which provides complementary information to each other for network prediction.

“time-extended internal organisation” → “temporal interaction pattern”,
“static closure graph structure” → “time-aggregated graph”.

Comment - [Minor, technical questions]

1. Lines 123-124. It is stated that the branching coefficients serve a “soft” rather than sharp censoring indicator. I do not understand what it implies and further explanation is needed.

Response - By “soft (sharp)” we mean a probabilistic (deterministic) indicator. In the literature of statistical learning, several assignment methods are at heart of many clustering algorithms, including the “hard (sharp)” assignments used by K-means and the “soft” assignments used by Expectation-Maximisation (EM) algorithms. In particular, our situation using branching coefficients is categorised in to the soft assignment method. Put simply, in the case of observable diffusion trees, the branching coefficients become a binary indicator being either zero or one, which corresponds to a “hard” assignment of the underlying links to branching set and chord set; in our problem settings of unobservable diffusion trees, a more reasonable choice is instead the “soft” assignment method from a probabilistic viewpoint. Note that applying the “hard” assignment method to our inferential scheme is equivalent to considering only one most probable diffusion tree for inference of the STN model, rather than being averaged over all possible diffusion trees, in which case the “hard” assignment method fails to establish the self-consistency condition for the iterative WTD estimator $\hat{\rho}$. It is for this reason that we have adopted the “soft” branching coefficients in our study.

Besides, we have used the concept of “censorship” borrowed from Survival Analysis (see Ref. [36] in the main text)—which refers to mechanisms for incomplete observations when analysing a variety of time-to-event data. In our situations, by “censoring” we mean an indicator whether the corresponding link is a chord or a branch during an observed diffusion cascade, noting that whether the waiting-time τ_{uv} occurred on link (u, v) can be observed (in this case of branches, τ_{uv} is read off as the nodal TDOA, $d_{uv} = t_v - t_u$) depends on if the diffusion occurred along this link is the earliest to arrive at node v , or otherwise “censored” by other adjacent links competing for spreading the information to node v .

Comment - 2. Line 152. The symbol “N” is used to represent a temporal network, however, I do not agree with this idea because “N” conventionally represents the number of nodes (the size of V in the terminology of this paper). It should be considered to replace it with other suitable symbols to avoid unnecessary confusion of readers.

Response - Thank you for the suggestion. To avoid notational confusion, we have adopted calligraphic letters to indicate sets and network-related terminologies throughout the revised manuscript, including “ $N(V, E)$ ” → “ $\mathcal{N}(\mathcal{V}, \mathcal{E})$ ”, “ N_s ” → “ \mathcal{N}_s ”, “ G ” → “ \mathcal{G} ”, “ T ” → “ \mathcal{T} ”, “ I_v ” → “ \mathcal{I}_v ” and “ J ” → “ \mathcal{J} ”. Also see Supplementary Table S5 for details.

Comment - 3. Line 157. Strictly speaking, the temporal event (u, v, t) is not in N but in E , and topological link (u, v) is not in G but in the set of edges of G . Such mathematical expressions should be carefully revised.

Response - Corrected [“(u, v, t) $\in N$ ” → “(u, v, t) $\in E$ ”]. We have carefully checked the mathematical expressions in our manuscript. In the literature of graph theory, it is often notationally convenient to ignore the distinction between a graph and its edge set. Adopting this convention, we thus do not use a new letter for G ’s edge set.

Comment - 4. Line 170. I do not get what the subscript “n” stands for. If “n” represents the number of events comprising the time-respecting path, then the set considered should not be E but E^n .

Response - Yes, the subscript denotes path length. We have revised the notation in accordance with that used for unicast path length. [“ $\{(s^*, v_1, t_1), (v_1, v_2, t_2), \dots, (v_{n-1}, v_n, t_n)\} \subset E$ with $t_1 < \dots < t_n$ ” \rightarrow “ $\{(s^*, v_1, t_1), (v_1, v_2, t_2), \dots, (v_{l-1}, v_l, t_l)\} \subset E$ with $t_1 < \dots < t_l$ ”]

Comment - 5. Lines 190-191. Is T actually tree or a directed acyclic graph with some loop if we discard the link directions? If the latter holds true, calling it a tree is confusing.

Response - In the case of continuous-variable WTDs, T is actually a tree, and the latter holds true for discrete-variable WTDs. In the main text, we assume that waiting-times are continuous, so we call T an s^* -rooted diffusion tree, while the discrete-variable case is discussed in Supplementary Note 9.

Comment - 6. Lines 203-204. I do not understand why L^{in} is a triangular matrix up to similarity permutation. Is G supposed to be a directed acyclic graph?

Response - This holds for arbitrary graphs. From the definition of the weighted in-degree Laplacian matrix L^{in} with off-diagonal entry $l_{ij}(D|N_s) = -\mathbb{1}_{[(j,i) \in G]} \lambda_{ji}(d_{ji})$, the corresponding entry vanishes if $d_{ji} = t_i - t_j < 0$, meaning that the DAT at node i is earlier than node j . Thus rearranging the network nodes in an ascending order of their DATs obtains a triangular matrix similar to Laplacian L^{in} .

Comment - 7. Line 230-231. It is stated that the Gibbs sampler iteratively flips any links. Is this mean to switch the presence and absence of a link in G ?

Response - Yes. For detailed implementation see Algorithm 5 in Supplementary Note 8.

Comment - 8. Line 265. The waiting-time distributions considered are shown in Fig. 3. I am concerning that all the distributions have a narrow support (effectively) in $0 < \tau < 4$. In reality, the waiting time distributions usually possess long tails which has considerable impacts on dynamical processes on temporal networks. Therefore, I want to see how the proposed method works when the waiting-time distribution follows a skewed distributions.

Response - The newly added simulations provide the inferential performance of applying our method to several empirical temporal networks characterised by their heavy-tailed WTDs. Numerical results show that our method still works even when the underlying WTD is spread over a rather wide support.

Comment - 9. Lines 269-270. The figure S6 indeed looks similar to a first-order phase transition, but further statistical-physics analysis is needed if it is said with absolute certainty. Otherwise the expression should be removed or weakened.

Response - We agree with this point of view. In light of the small network size we used in our benchmark tests, we have weakened our expression (“first-order phase transition” \rightarrow “phase transition”).

Comment - 10. Lines 276-277. I do not get why the adoptive mechanism for the fitting explains the smaller number of cascades required for inference than expected. Further explanation is needed.

Response - To explain this point, we first note the key observation behind Glivenko-Cantelli Theorem⁵ that the estimate error decays with sample size of available data. In contrary to the asymptotic case of large sample size, when there observed a substantial distributional divergence between the postulated real WTD and the realised WTD (which could be the case especially when only a few diffusion data are at hand), the inference using the “actual” WTD $\hat{\rho}$ usually outperforms using the “given” WTD ρ . This is because through the process of inference, the former acquires subtle distributional features of the realised waiting-times in the given diffusion cascades. In other words, simultaneous reconstruction of G and ρ can obtain better inferential performance⁶ than sole topology inference due to the usage of estimated WTD $\hat{\rho}$ that is inferred from data and thus “adaptive” to the particular realizations of diffusion cascades.

Another, more straightforward, way is to imagine the following scenario: the underlying WTD $\rho(\tau)$ of the model is assumed to be Gaussian with centre $\mu = 1$, but despite an extremely low occurrence probability, actual waiting-times during the observed cascades all fall into the interval $\tau \in (0.5, 0.9)$. In this fictive example, it would be better to perform inference using the data-driven WTD estimate centred at $\hat{\mu} = 0.7$ than the true (but erroneous for this extreme example of the particular realizations) Gaussian distribution centred at $\mu = 1$.

Comment - 11. Line 325-326. It is stated that the reconstructed STN gives statistics sufficient for reliable prediction, which I do not agree. It is shown in the manuscript that the reconstruction methods performs well for the STN benchmarks, however, no prediction is made through this procedure. Thus, additional evaluation should be provided to demonstrate the predictive power of the proposed method.

Response - As discussed in our previous response (see page 2), the STN model is of both theoretical and practical interest due to its simplicity and mean-field nature. Remarkably, a number of analytical methods (see, e.g., Refs. [16, 25, 33, 34] in the main text) have in recent years been developed for quantifying diffusion dynamics on networks based on the STN model or on the mathematically equivalent models. Thus the STN model’s predictive power have already been justified by the existing literature. Applying such analytical methods to the inferred STNs is mere repetition of the previous works discussing a variety of forward problems. Please see our response to the second reviewer (page 12) for discussion on the Hawkes process method. Returning to the previous metaphor (refer to the second part of our response, page 3), in light of many known physical laws expressible in terms of “temperature variable” (=extant analytical tools based on the STN framework), the ultimate goal of our manuscript is then to offer a method to gauge the “temperature” (=STN null model) of a temporal network, instead of its molecular-motion-level details (=individual-level temporal interactions).

Comment - [Comments on figures and tables]

Fig 1. The panels (a)-(d) are difficult to see as they are distorted. Alternative way of their presentation is recommended.

⁵The Glivenko-Cantelli Theorem states the asymptotic properties of an empirical distribution function as the number of i.i.d. sampled data points grows. We have used this result as a key step in the proof of our WTD estimator’s asymptotic consistency showing that $\hat{\rho} \rightarrow \rho, C \rightarrow \infty$ (see Supplementary Note 4).

⁶Interestingly, such effects are also present in many supervised learning tasks, which usually act, however, a negative role. More concretely, due to the process of training, the model or model parameter(s) learned from data similarly acquires subtle correlations with the particular realization of training inputs, thus causing the so-called overfitting problem. This leads to a reduced predictive capability, meaning a much worse prediction performance for the held-out testing data than that for the training data.

Response - We have replotted Fig 1 with a toy network in a more illustrative style. Please also see our response to the relevant point raised by the third reviewer, page 15.

Comment - Fig 2. In caption, the notions of “incorrectly existing links” and “missing links” are not defined and hard to understand. I suggest to replace them with more common notions like “false-positive links” and “false-negative links”.

Response - Replaced. Thank you for this suggestion. We have also corrected other relevant notions, e.g., “actually existing links” → “true-positive links”.

Comment - Fig 4. It seems that the panels (a)-(c) are not mentioned in the main text. If so, these panels should be removed. If they should be kept, at least the legends of them should be enlarged to be readable.

Response - We have moved Fig 4(a)-(c) to the Supplementary Information, the size of which has also been enlarged.

Comment - Table 1. I do not get the idea of the right-bottom element of each cell, “simultaneous reconstruction of both underlying networks and associated WTDs”. Is that what was done throughout this study, isn’t it?

Response - In our numerical study we distinguish the following two cases: in the first, the real WTD is supposed to be explicitly given to estimate the STN model, thus only the topology inference for time-aggregated graphs being implemented; in the second (to which case the right-bottom entries correspond), the underlying WTD is unknown and also to be inferred only from observed diffusion data, together with the time-aggregated graph, by implementing the iterative coordinate-ascent inference procedure.

Comment - In addition, relative sample size is mentioned in the caption but not presented in the table.

Response - Here the critical sample size reported in Table 1 should be “critical relative sample size”. We have carefully checked the figure and table captions to remedy such typos.

Comment - Table 2. Should be provided the exact correlation coefficient values for these factors, beyond their tendency of positive or negative correlations.

Response - We have carried out extensive numerical simulations to quantify and visualise such correlations between inferential complexity of STN models and their structural and temporal ingredients, G and ρ , respectively. (See Supplementary Figure S27–S30.) For the structural aspects, We have mainly focused on three types of time-aggregated networks, Erdős-Rényi random graphs, Watts-Strogatz small-world and Price scale-free networks, with parametrically tunable clustering coefficients, average path lengths and/or power-law exponentials of node degree distributions. The sizes of the networks we used in our numerical experiments are also increased to $N = 1000$, ensuring that the scale-free networks display substantial heterogeneity in node connectivities for comparison with random or small-world graphs. Differing from our previous observation from a few benchmark time-aggregated networks, we find that there exist obviously non-monotonic dependence between

structural parameters and inferential complexity, rather than being linearly correlated. For example, the curves of inferential complexity versus both clustering coefficient and average path length show significant turnover trends. In light of this non-linearity of dependence, we have not have removed Table 2 from our manuscript to avoid misleading conclusions on from a limited set of benchmark networks. However, we note here that in the practically relevant regime of salient small-world effects (namely, large clustering coefficient and small average path length, as many real-life networks exhibit), the inferential complexity is positively (negatively) correlated with clustering coefficient (average path length) of the time-aggregated network, which agrees with our previous benchmark test results reported in Table 2 of the original manuscript. For the temporal aspects, we examined the roles of the first three (normalised central) moments—the mean, variance and skewness—of underlying WTDs, which also coincide with the previous benchmark tests. Here we also mainly focused on the most practically relevant WTD, Pareto distribution with varying power-law exponentials as well as different lower bounds of density support. In particular, we have further considered the effect of minimal waiting times in affecting the inferential complexity, partially in consideration of the role of minimal interevent times in non-Poissonian diffusion dynamics as recently reported in [9]. The relevant numerical results for inferential complexity of temporal networks can be found in the revised Supplementary Information.

- [9] Jo, H. H., Perotti, J. I., Kaski, K., & Kertsz, J. (2014). Analytically solvable model of spreading dynamics with non-Poissonian processes. *Physical Review X*, 4(1), 011041.

Responses to the Second Reviewer

Comment - Dear Editor,

In this paper, the authors focus on the important problem of reconstructing structural and temporal patterns from empirical observations in networked systems. To do so, the authors adopt the perspective of renewal processes, or stochastic temporal networks, allowing to take into account the heterogeneous inter-event times observed in empirical systems, while imposing some statistical constraints. This work brings without hesitation an important contribution to the active field of temporal networks. The authors mostly focus on situations where adjacent edges are independent, but also describe the situation where correlations are non-negligible. I am overall very positive about this work, clearly written, and whose theoretical contribution is sound and consequent. However, I am less convinced by the empirical validation, mainly performed on artificial data. If possible, I would advise the authors to test the method on real-world temporal networks, where some of the modelling assumptions are not verified (e.g. non-stationarity). The authors should also draw connections with different works adopting a different, yet related approach for network reconstruction and cascade size predictions, using Hawkes processes (very similar to random walks on stochastic temporal networks). Examples include:

<http://snap.stanford.edu/seismic/seismic.pdf>

<https://www.mpi-sws.org/manuelgr/pubs/S2050124214000034a.pdf>

After these minor points are addressed, I will be very happy to recommend the publication of this work in *Nature Communications*.

Response - Thank you for your positive assessment of our paper. We have taken your advice to validate our method using empirical temporal networks (which was also put forward by the editor and other reviewers, see, e.g., our relevant responses to the third reviewer, page 15). Also, thank you very much for the pointer to the interesting works on Hawkes processes. We have discussed the connection of our work to the Hawkes process approach (see the revised Discussion section).

In addition, we have tested our null model assumptions on empirical temporal networks. As pointed out many times (see, e.g., pages 2 and 3), the STN model would usually be thought of as an oversimplification of the reality. This is indeed the case, but it is a compromise that allows discovery of latent network structure in data at an achievable accuracy in accordance with the volume of data collection. However, we still need to be aware of and to tell, possibly in a quantitative way, the difference between the real and predicted temporal networks. Specifically, we considered three statistical quantities with respect to empirical WTDs observed in real-world temporal networks—distributional standard deviation (DSD), Pearson correlation coefficient (PCC), and normalised mutual information (NMI)—to measure the extent to which the link-level waiting times of a temporal network are i.i.d. distributed as $\hat{\rho}$ of the fitted STN model from different perspectives (see Supplementary Note 10 for more details.)

Once again, we wish to emphasise here that empirical data show, unsurprisingly, that most of real-world temporal networks are far from the perfect regularity as our basic STN null model postulates, and their underlying WTDs are typically heterogeneously distributed and complicatedly correlated. In the language of statistical hypothesis testing (for example, applying Kolmogorov-Smirnov test to compare any two link-level empirical WTDs, ρ_{uv} and $\rho_{u'v'}$), we should reject the null hypothesis (namely, the STN model's assumption that $\rho_{uv} \equiv \rho_{u'v'}$) at a given significance level. However, as is already mentioned (see page 2), the STN model aims at capturing such regularity of latent tempo-

ral networks with tunable approximation, no matter how complicated they may be. Although this marked deviation of real-world temporal networks from the STN model causes a sensibly decreased inferential ability (see Table 3 in the main text), our method still provides the inferable structural and temporal statics of underlying temporal networks with the corresponding accuracy. Of course, the higher accuracy of reconstructed networks from the fewer data of diffusion process observations, the better inferential power of our method.

Keeping in mind this deviation of STN null models from reality, a consequent question arises: How approximate is the reconstructed STN as compared to the real network on which diffusion took place? This question should be of equal and even greater importance concerning the confidence level analysis for the estimated STN models. Fortunately, many sophisticated statistical techniques are adaptable to our problem settings. Here we adopted the bootstrap method—originally proposed by Efron [10] and standardised as a computer-intensive resampling procedure for hypothesis tests and confidence interval estimation [11]—for calculating the confidence bands of estimated WTDs. Note that resampling DAT data from observed diffusion cascades is a formidable task, partially due to the non-Markovianity of multicast processes—the diffusion arrival depends on the a set of competing events from the history of the diffusion dynamics, unlike the case of unicast diffusion (such as random walks and, more generally, Markov chains) with “interchangeable” trajectories⁷. Our preliminary confidence analysis are hence based on the parametric bootstrap procedure, where both G and ρ are regarded as generalised “parameters” of the data generative process (also see the footnote on page 3), and the bootstrap resamples of diffusion cascades can thus be generated using the corresponding STN tuple $\hat{N}_s = (\hat{G}, \hat{\rho})$ estimated from the original diffusion data. Detailed algorithm descriptions and benchmark test results are provided in the revised Supplementary Information. There are many other conceivable resampling strategies, and/or statistical tools (e.g., cross-validation techniques) for accessing the confidence level of the estimated STN model. These issues deserve future exploration.

[10] Efron, B. (1979). Bootstrap methods: Another look at the jackknife. *Annals of Statistics*, 7, 1-26.

[11] Efron, B., & Tibshirani, R. J. (1993). An introduction to the bootstrap. *Chapman and Hall, CRC Press*.

[12] Felsenstein, J. (1985). Confidence limits on phylogenies: An approach using the bootstrap. *Evolution*, 783-791.

[13] Felenstein, J. (2004). Inferring phylogenies. *Sunderland: Sinauer Associates*.

⁷The interchangeability of unicast trajectories means that for two observed paths $J = \{v_1, \dots, v_l\}$ and $J' = \{v'_1, \dots, v'_l\}$ intersected at node $v_k = v'_{k'}$, then the shuffled paths $\tilde{J} = \{v_1, \dots, v_k, v'_{k'+1}, \dots, v'_l\}$ and $\tilde{J}' = \{v'_1, \dots, v'_{k'}, v_{k+1}, \dots, v_l\}$ are still legal as reasonable bootstrap trajectories. It is noteworthy that many brilliant methodologies for resampling sequential data were originated from phylogenetics [12], aiming at computation of the confidence level of phylogenetic trees from bootstrap samples of, e.g., species' DNA sequences [13]. This is an apparently different problem but deeply related to ours, noting that the TDOA information is able to be encoded in sequence distances, and equivalently, the phylogenetic tree is the inferred (acyclic) network from DAT data. Unfortunately, these bootstrap-based tools for analysing confidence levels cannot be directed applied to our problem setting, mainly due to this acyclicity condition that holds for phylogenetic trees but not for general graphs. The multicast nature of underlying diffusion processes breaks the interchangeability of DAT cascades, and finding statistically meaningful bootstrap resampling methods is an important but challenging task. Here we adopt the parametric bootstrap procedure (see below) using the inferred STN model, leaving its non-parametric counterpart as an open question.

Responses to the Third Reviewer

Comment - The paper studies the problem of reverse-engineering the structure of temporal networks when detailed temporal data on dyadic events is missing. It develops an efficient algorithm for doing so, using a stochastic null model of diffusive arrival times, assuming no dependence between different dyadic interactions. Benchmark tests applied to synthetically generated networks and to toy time-aggregated networks show that temporal networks can be unveiled with high accuracy.

I think the paper is well researched: it cites all the recent temporal network literature and its applications of which I am aware adequately, and makes good points why the problem of reconstructing temporal networks is important, such as in epidemiology. Clearly there are broad applications to the problem. Concerning language, the paper is also well written. It is good that the case of correlations of events between different dyads is discussed, as this would have been my first objection in terms of limitations. As having not the most technical expertise in this specific sub-field, I was not able to follow all the technical derivations, but from what I can tell they seem correct.

Response - Thank you very much for your positive comments.

Comment - The title and promise of the paper is very interesting, however I had 2 major problems with the manuscript. These might be problems that can be fixed, but it is not clear to me as the lack of overall clarity is one part of the problem.

One issue concerns only the presentation and outline of the big picture. A problem for me was the low readability of the Results section due to the extreme technicality of the subject. In other words, the “Why are we doing this?” and the “So what?” questions often remain unanswered, which I believe should not be the case in a non-specialist journal. For example, I can see the different steps that build up the algorithm, but why are they important? The sub-sections read “We next calculate.., We now introduce.., This step takes.., This step takes..” without saying “This is important because..”. I understand that most of the mathematical derivations have been deferred to the SI, which is good, but still, I feel the manuscript’s main results remain intractable to the general reader. Certainly I would not expect a general reader to understand everything in the technical development of such a complex algorithm, but somebody from the network science community should at least be able to get half of it. I doubt this is the case in the current presentation. I am sorry my criticism is so vague, but the vagueness comes from not fully understanding from the manuscript the importance of what the steps of the model do to advance previous approaches or the field itself. For example, maybe the point on privacy could have been elaborated better.

Response - First of all, we fully understand and completely accept the criticism as invaluable feedback. We have been aware that the impression a reader forms about our problem setting and/or algorithm structure might be quite unclear, primarily because the original manuscript was written tacitly (but wrongly) assuming that the readers have a fair amount of statistical expertise at the network inference literature. Consequently such technical hurdles lower the attractiveness of our manuscript to a wider spectrum of audience. Thank you very much for pointing it out. In fact, it is to a substantial extent overlapped with the points raised by the first reviewer (see, e.g., page 2 for explanation of questions like “why doing this”, “so what”, etc.; see page 4 for elaboration of the privacy issue). We have attempted to enhance the readability of our manuscript, giving more explanatory descriptions of our methodology with illustrative figures (see next response on page 15), and avoiding unnecessary ter-

minologies (see page 6 for our response to “Terminology”). We hope that we have answered your comments, at least partially, and that the revised manuscript has an enhanced readability now.

Comment - Minor issues on clarity are also present in the figures. For example, the toy network used in Figure 1 is not so simple, and the concepts it aims to explain are lost because of that. In this figure, the ROC curves are shown, but the text or the caption do not deliver the message of what this actually means or why it is important. It is not clear what the legend refers to (#1, #2,...).

Response - Yes, we have also recognised this problematic issue. In the revised manuscript, Figure 1 has been replotted with a small-size, toy network in order to better illustrate our inferential scheme. Besides, the ROC curves have been removed from the updated figure, because they are unimportant for the purposes of clarifying our basic concepts.

Comment - The second issue I see is the lack of tests with relevant (large) data sets from real systems. In a specialised theory-oriented journal, the vast array of benchmarks and small, time-aggregated toy networks which were tested, would be more than enough. However, in an important general paper on reconstructing temporal network structure, I would expect the test of the proposed model on a real data set where detailed link events are available, and comparison of the reconstruction results to the case where the time information was aggregated. This data could be mobile phone calls, internet messages/emails, or any of the large-scale temporal data sets available online, for example from <http://www.sociopatterns.org/datasets/> or from <http://projects.csail.mit.edu/dnd/>. These are just first results from a quick internet search, I am sure there are more data sets available publicly.

Response - Thank you very much for your suggestion. Following that, we have applied our method to several empirical temporal networks (with small and intermediate size) to test its validity under more realistic situations (also see our relevant response to the second reviewer, page 12). We have found many large-scale temporal datasets available at the above websites, among others, but nevertheless we have to abandon using such large empirical networks primarily due to the non-scalability of our algorithm⁸. Note that the sample size of diffusion cascades used for inference is of the same order as the network size, the time complexity of our method is actually $O(N^3)$. Additionally, the two basic steps—MCMC sampling from the space of possible network configurations and self-consistency iterations in functional estimation of WTDs—may contribute a potentially large factor making the inferential procedure extremely time-consuming, despite not affecting the polynomial order of complexity. We hope you understand that we cannot, at least at current stage, fulfil the empirical validation on the extremely large-scale realistic networks, considering this complexity arising from the intrinsic nature of such a time-augmented inverse problem we considered here. Next we would like to give some further discussion on possible speedups of our algorithm for future research.

First, as shown in Supplementary Note 3, the step of topology inference is divided into well-paralleled subproblems, each solving the maximum-likelihood estimation of the local configuration I_v centred at node v independently, implying that a scalable procedure for topology inference is possible using parallel programming. In fact, this has been already utilised in many previous works on network inference using convex optimization techniques (see, for example, [4, 10]).

⁸It is widely recognised that the scalability of algorithms—meaning the applicability to large-scale data—is becoming increasingly important in the era of Big Data. In network science, algorithms with quadratic (or worse) computational complexity are typically thought of as non-scalable to large-scale networks have, e.g., millions of nodes [10].

Second, the number M of MCMC iterations used in topology inference determines different precisions (in predicting *a posteriori* presence probabilities of links), which is also the main reason for an intolerably high coefficient of computational complexity. Also, the Gibbs sampler we adopted for learning the time-aggregated links is quadratic to the node size (or, equivalently, N^2 hypothesis tests to be performed), and hence not scalable to large-scale networks. Therefore new operative procedures are needed to improve the order of complexity. Successful examples include the dynamic message-passing technique based algorithms to infer the origin [11], propagation paths [12], and design immunization strategies [13] for the epidemic dynamics on (static) networks. However, we failed to use such more efficient methods mainly because of the following two differences in our problem settings: (a) the relaxed assumption of locally tree-like network structure and (b) the increased volume of parameter search space. Here we exclude the unrealistic (especially for social networks) assumption that the diffusion network has a locally tree-like structure, while the message-passing or other belief-propagation methods are often problematic in convergence for loopy graphs. Also, note that the number of possible network configurations is overwhelmingly larger than that of possible origins (or epidemic pathways) embedded in a given network, let alone in combination with the functional space of WTDs consisting of arbitrary shaped distributions making the design of message-passing mechanisms rather difficult, if not absolutely impossible. As we have discussed in the manuscript, we are of the view that the link-level estimation can be achieved by a scalable procedure due to its voluminous search space, except that we zoom to a coarse topological scale to find an appropriate trade-off between the prediction accuracy and computation complexity.

Finally, similar to the role played by the MCMC iteration number M in topology inference, the number of discretisation bins affects both inferential precision and algorithmic complexity of the WTD estimator. Straightforwardly, the implementation with bin width δt has twice number of parameters to be learned (i.e., the probability masses $\rho[i]\delta t = \int_{i\delta t}^{(i+1)\delta t} \rho(\tau')d\tau', i = 0, 1, \dots, \lceil l_\tau/\delta t \rceil$) and thus about four times computation complexity than the WTD estimator with bin width $\delta t' = 2\delta t$. Motivated by this, a promising speedup of our algorithm by reducing the parameter space is to perform the transform from time-domain estimation to Fourier-domain. (see Supplementary Note 9 for details). Mathematically, $\rho[i]$ is also rewritten as $\langle \rho, b_i \rangle = \int_0^{l_\tau} \rho(\tau')b_i(\tau')d\tau'$, the inner product of WTD $\rho(\tau)$ and the i -th standard orthogonal basis $b_i(\tau) = (1/\delta t)\mathbb{1}_{[\tau \in [i\delta t, (i+1)\delta t])}$ of the “binning” operation on WTDs. Therefore the binned WTD $\rho[i]$ also constitutes a generalized Fourier series of $\rho(\tau)$. In fact, our inferential framework is readily extended to any generalized Fourier series representation, ranging from Fourier-Bessel or other orthogonal polynomial series expansions to wavelet domain. A consequent question is which series performs the best for the purpose of density estimation. Here we leave as an open problem the optimal orthogonal series estimates that minimise the predictive error, in particular for the empirically relevant power-law distributions exhibiting marked anomalous patterns such as bursts and heavy tails.

- [10] Du, N., Song, L., Gomez-Rodriguez, M., & Zha, H. (2013). Scalable influence estimation in continuous-time diffusion networks. *Advances in Neural Information Processing Systems*. 3147–3155.
- [11] Lokhov, A. Y., Mézard, M., Ohta, H., & Zdeborová, L. (2014). Inferring the origin of an epidemic with a dynamic message-passing algorithm. *Physical Review E*, 90(1), 012801.
- [12] Massucci, F. A., Wheeler, J., Beltrán-Debón, R., Joven, J., Sales-Pardo, M., & Guimer, R. (2016). Inferring propagation paths for sparsely observed perturbations on complex networks. *Science advances*, 2(10), e1501638.
- [13] Altarelli, F., Braunstein, A., Dall’Asta, L., Wakeling, J. R., & Zecchina, R. (2014). Containing epidemic outbreaks by message-passing techniques. *Physical Review X*, 4(2), 021024.

Comment - In summary, the authors have clearly put a lot of effort into the mathematics and seem to have done a great job with the technical details, but the main results and their consequences are too much hidden behind the technical nature of the presentation which makes it hard to judge the paper's impact.

Response - Thank you for your evaluation. All of the comments raised by the editor and reviewers has been carefully addressed in our revision process, and improving the presentation of our manuscript has also been one of our main endeavours. We hope it achieves a satisfactory level of readability now. Finally, we provide a brief summary of changes below.

Once again, we sincerely appreciate all of your help and very constructive comments on our manuscript, without which our paper could not have evolved from the original draft to its current state.

Summary of Changes

- Figure 1 has been replotted with a small-size network to better illustrate our inferential framework.
- The data-privacy issue has been elaborated to better motivate our study.
- The role of WTD estimator in the adaptive fitting of diffusion data has been further explained.
- The confidence band for WTD estimation has been analysed using parametric bootstrap method (see Algorithm 7 in Supplementary Note 8 for detailed implementation and Supplementary Fig. S7 for benchmark test results).
- Our method has been examined on several realistic temporal networks (See Supplementary Table S3). Empirical validation results (see Table 2 and Supplementary Figs. 22–23), as well as relevant discussions have been provided.
- Three statistical measures (DSD, PCC and NMI) has been introduced to quantify the deviation of empirical datasets from the null model assumption (see Table 2 and Supplementary Note 10).
- The Fourier-domain estimation has been considered to reduce the parameter space for better self-consistent convergence properties of our WTD estimator (see our relevant discussion on empirical validation results and Supplementary Note 9).
- Quantitative correlations between the inferential complexity $\xi(D, \mathcal{N}_s)$ and structural (or temporal) properties of STNs have been numerically examined (see Supplementary Figs. 27–30) and further discussed.
- The connection of our work with the Hawkes process method has been discussed.
- Following the reviewer’s comment, Fig. 4(a)-(c) have been enlarged and moved to Supplementary Figs. 24–26.
- For the length limit, the Introduction section has been shortened, and some paragraphs have been moved from Supplementary Note 1.
- Part of the notations used in our manuscript have been changed based on the reviewer’s comments. See Supplementary Table S5 for summary.
- Some minor revisions have been made to the manuscript following the detailed suggestions of the reviewers. Please refer to our point-to-point responses to the relevant comments.
- Other revisions have been made to meet the editorial criteria of Nature Communication (e.g., word counts, font styles, etc.) All changes have been highlighted in the revised manuscript text.

Reviewer #1 (Remarks to the Author):

The authors addressed the issues that I raised in the previous report. The revisions resolved some of them, but the main issue still remains. Therefore, I would like to ask the authors for further revisions of the manuscript after going through the following arguments.

1. Importance of the inference of the stochastic temporal network (STN) model.

In the response, the authors stated that the inference of the STN model is of interest because of its flexibility and goodness as a null model of empirical temporal networks.

I agree that the flexibility of the STN model is its advantage, however, I do not agree its goodness as a null model of empirical temporal networks. My disagreement is based on the results of the inference on empirical data sets shown in Table 2. If I understand correctly, the column labeled with "BEP_1" indicates the precision (recall) measure at the break-even point in the case the waiting-time distribution and the aggregate network are simultaneously estimated based on the simulation of cascades on the empirical contact sequence. The precision measure of BEP_1 are in the range from 0.20 to 0.36 for various data sets. I believe that these precision values are too low to claim the goodness of the model.

In the response, the authors also referred to the motivation of this study as:

"... there is lack of appropriate methodologies for assessing whether and which "parameters" of the STN model $N = (G, \rho)$ fits empirical data."

In my opinion, there have been few attempts to fit the STN model to empirical data simply because the research community notices that it is too far from reality.

If I can borrow an example from static network analysis, it seems like trying to fit Erdos-Renyi random graph model to empirical network data. I have no doubt about the theoretical and historical importance of the ER random graph. However, I never expect the goodness of its fit to empirical networks because real-world networks are probably more heterogeneous and well-structured than the ER random graph. In a similar way as this, the research community respects the theoretical importance of the STN model, but does not regard it as a good model of reality, on the basis of numerous studies about the different kinds of correlations observed in empirical temporal networks and their impacts on diffusion dynamics.

The failure to accurately work on empirical temporal networks makes the inference of the STN model not convincing to general readers as a practical tool.

2. Distinction between public and private information.

Thanks to the examples that the authors provided in the response, now I understand what the authors meant by "public" and "private" information. I recommend to rephrase them by "observable" and "unobservable" information, respectively. In the example of viral marketing, the first-arrival-time data is available only to companies, not public, and is definitely containing private information of customers. It requires no argument that the data of "who buys what at when" is very sensitive and private information. In the example of blogspace, indeed one can say that blog posts are public and interactions between bloggers are private. However, I suggest to consider to use "observable" and "unobservable" to explain these two examples in a consistent way. Although my concern about this point might seem irrelevant to science, I strongly believe that this point is really crucial to clarify the core of the motivation of this study.

3. Simulations on empirical temporal networks.

The proposed method assumes that any node u has finite t_u , i.e., the diffusion reaches all nodes in the network in a finite time. However, in empirical temporal networks, not all of node pairs are reachable through time-respecting paths (e.g., see Table 2 in Starnini et al., Physical Review E 85, 056115 (2012)). Was this assumption approved for the data sets shown in Table 2? If not, how can the discrepancy be justified?

In Table 2, it is almost impossible to understand the difference between the four experimental setting that led to BEP1-4. Furthermore, the BEP3 and BEP4 are never mentioned in the main text. The experimental settings should be clearly described.

Reviewer #2 (Remarks to the Author):

The authors have thoroughly addressed my comments. I am now in favour of the publication of this manuscript in Nature Communications.

Reviewer #3 (Remarks to the Author):

I thank the authors for the detailed revision and rebuttal. I think the readability was improved, such as with more examples on applications in marketing and a much better to understand toy network in Fig 1, which is now also comprehensible to a non-expert reader. The paper is still quite technical, but this is probably hard to avoid with such a topic. I am still a bit unsure how attractive the paper would be to a general audience, but definitely more than in the manuscript's first version. I have no concrete further criticism to add in this regard.

Concerning the size of the empirical networks, I completely understand if the current algorithm cannot handle large networks. This is of course not optimal but a good first step. I think the use of several small empirical networks is fine. The case of higher-order correlations was now discussed even further.

Again, my expertise is too limited to decide whether the STN model is realistic or impactful enough, see the first reviewer's main objection. Certainly, in the last years many forms of human behavior have shown to be bursty with heavy-tailed WTDs, but I can also see scenarios of recurring human behavior, such as daily routines in online or mobility behavior, where WTDs are homogeneous and possibly covered by an STN model.

In conclusion, I am mostly positive but also slightly in doubt. However, the technical quality of the manuscript seems high and could strongly influence the thinking in the field, so I see no good reason to not publish it.

Responses to the First Reviewer

Comment - The authors addressed the issues that I raised in the previous report. The revisions resolved some of them, but the main issue still remains. Therefore, I would like to ask the authors for further revisions of the manuscript after going through the following arguments.

1. Importance of the inference of the stochastic temporal network (STN) model.

In the response, the authors stated that the inference of the STN model is of interest because of its flexibility and goodness as a null model of empirical temporal networks.

I agree that the flexibility of the STN model is its advantage, however, I do not agree its goodness as a null model of empirical temporal networks. My disagreement is based on the results of the inference on empirical data sets shown in Table 2. If I understand correctly, the column labeled with “BEP₁” indicates the precision (recall) measure at the break-even point in the case the waiting-time distribution and the aggregate network are simultaneously estimated based on the simulation of cascades on the empirical contact sequence. The precision measure of BEP₁ are in the range from 0.20 to 0.36 for various data sets. I believe that these precision values are too low to claim the goodness of the model.

Response - Thank you for the positive comment on the flexibility of our STN model. Regarding the goodness-of-fit, the obtained BEP₁ is admittedly low, reflecting the inaccuracy of our model in fitting real-world temporal networks under the specific experimental setting (as correctly understood). We also agree with you that these precision values of BEP₁ at such an unsatisfactory level (≤ 0.36) never support the goodness of STNs used as a model that aims at explaining empirical network data. The expected solution is, of course, to advance the reconstruction accuracy of our methodology, but we find that there is an intrinsic limit to further improve BEP₁. In the following, we first explain this underlying algorithmic limitation, and then distinguish between the roles played by the STN model in *modelling* and *null modelling* of temporal networks. We show that, despite being far from optimal efficiency for the purpose of network modelling, inference of the STN model is of particular statistical importance in the sense of null modelling. Hopefully that our response will be satisfactory to further clarify this remaining issue.

1. Methodological causes leading to low BEP₁ and beyond.

We note first that under the specific problem setting in which both G and ρ need to be simultaneously estimated from empirical diffusion cascades \mathbf{D} , the coordinate-ascent iterations are required. The ultimately converged STN estimate $\hat{N}_s = (\hat{G}, \hat{\rho})$ —self-consistently obtained in topology inference and density estimation steps—will be susceptible to the extent that a real temporal network departs from the STN model assumptions. This is due to a positive feedback effect leading to the reinforced error between the consequent updating of $\hat{\rho}^{k+1}(\hat{G}^k, \mathbf{D})$ [$\hat{G}^{k+1}(\hat{\rho}^k, \mathbf{D})$] and the former estimated output \hat{G}^k [$\hat{\rho}^k$]. As a result, the small values of BEP₁ show that the iterative coordinate-ascent procedure fails to converge to the true (mean-field) WTD, and hence gives erroneous topology inference results.

To avoid such unexpected feedback loops, we performed comparative numerical validation on precision of the estimated network $\hat{G}(\rho, \mathbf{D})$ using real WTD ρ (see BEP₃ in Table 2), showing an acceptable efficacy of our topology inference method even when applied to real-life temporal networks. Here, ρ is the population-wide, empirical distribution of waiting times that are read off from realistic contact sequences. Similarly, given the real time-aggregated network G as algorithm input, we also obtained

the improved density estimation results $\hat{\rho}(G, \mathbf{D})$ for underlying WTDs (see blue dashed lines in Supplementary Figures 22–23). In summary, the observations above suggest that the proposed method for either topology inference or density estimation still works for empirical temporal networks, provided that extra temporal or structural information, together with observed diffusion data, is available.

As a concluding remark, when no other prior knowledge is given (as in the case of BEP_1), the reconstruction accuracy of our algorithm often collapses due to the inherent feedback loops in the iterative coordinate-ascent procedure that amplifies the estimation error caused by the unrealistic STN model assumptions¹. However, insomuch as BEP_1 does not convey the whole story, our method is usable in a variety of applications and hence of practical interest (also see Discussion in the main text). In fact, the prior information other than collective diffusion data (e.g., partially observed topology structure, relevant metadata of network agents, prior WTD estimated from small samples of temporal events, etc.) is practically available for network inference tasks, which readily enters our inferential framework in a Bayesian fashion and enhance the estimation performance promisingly (see Supplementary Note 7).

2. Distinguishment between *model* and *null model*.

As stated in the very beginning of our response, we also do not claim the goodness of STNs as a *model* of temporal networks in view of the reported BEP_1 , but statistically speaking, the STN model is nevertheless a *null model* of them—two distinct concepts which should be strictly distinguished from each other. As mentioned in the main text and previous responses, the added time dimension of temporal networks, embedded in form of the event set $\{(u_i, v_i, t_i)\}$, introduces too many degrees of freedom to allow a faithful recovery from measured data. To escape from this impasse, our inferential method based on the STN model is naturally motivated with the help of the null model assumptions—the objective of our work is thus *null modelling* of temporal networks, rather than *modelling* or explaining the empirical observations. Keeping in mind that the role of null models is by comparison with real systems to demonstrate some characteristics of interest they potentially possess, it is hence conceptually plausible that, in our situation, one uses the STN model for the purpose of investigating non-i.i.d. distributed interevent times (as well as their consequent effects) of real temporal networks. In fact, null models have been widely used in the empirical study of temporal networks [1], where shuffled event sequences are constructed in a variety of ways to systematically remove heterogeneities and/or correlations of original temporal interactions. It is such comparisons of these nullified networks with real-world networks that revealed a series of empirical findings (see, e.g., [2,3]) which, to a substantial extent, catalysed the birth of the temporal network theory. Similarly, the STN model also provides flexible levels of nullification of the effects arising from a complex temporal interaction pattern, which is therefore important in both empirical and theoretical study of temporal networks. In particular, our procedure also corresponds to a data-mining approach: from the reconstructed *null model* one can extract the statistical regularity or properties of temporal networks, which we believe is of even greater importance than a *model* that aims at fitting the empirical data accurately.

Furthermore, we note here that the values of BEP_1 reported in Table 2 only indicate the reconstruction performance, not the goodness of the STN model *per se*. Strictly speaking, BEP is just a measure-

¹In such regimes of empirical networks far from the STN model assumptions, the task of temporal network reconstruction *only* from observed diffusion data will be extremely difficult, if not absolutely impossible. The reason lies in the fact that agent heterogeneity and/or temporal interaction correlations dramatically alter dynamical properties of the networked diffusion. Another, perhaps more straightforward, reason is information-theoretic: the minimum required sampling rate can hardly be achieved in practice, were underlying WTDs so non-homogeneous that requires to be separately recovered from data.

ment index proposed for assessment of topology inference algorithms. Therefore small BEP_1 shows the inadequate extent to which empirical temporal networks can be inversely inferred under a specific experimental setting (for details see our response on page 6), but does not tell whether or how appropriately STNs can serve as a null model of temporal networks (see Introduction and Supplementary Note 1 for justification of our STN model used in the *null modelling*). To further explain this point, please allow us to borrow an example from static network analysis. The intuition behind the modularity maximisation techniques—a mainstream approach to detecting network communities—is to use stochastic blockmodels as null networks to quantify the community effect exhibited by a real network [4]. Here, a so-called network modularity plays a similar role as BEP_1 does in our problem setting. If the detected network modularity is quite low, then it may only show an unexpected performance of the community partitioning method, but no problematic issue on the definition of network modularity (and hence the justification of stochastic blockmodels as the null model of modular networks).

As a remark, we already considered the *modelling* accuracy of the STN model for temporal networks in the last round of revision, and instead of BEP_1 , we proposed three indices, DSD, PCC and NMI to measure the discrepancy between the STN model assumptions and empirical contact sequences (see Supplementary Note 10 for details). Numerical results show that realistic temporal interactions often demonstrate relatively weak heterogeneities and relatively strong correlations (see relevant discussion on Table 2). The values of such indices support that the STN model is also roughly acceptable as an approximate *model* of realistic temporal networks, noting that empirical features of temporal interactions can be further captured by higher-order STN models.

- [1] Holme, P. (2005). Network reachability of real-world contact sequences. *Physical Review E*, 71(4), 046119.
- [2] Pan, R. K., & Saramäki, J. (2011). Path lengths, correlations, and centrality in temporal networks. *Physical Review E*, 84(1), 016105.
- [3] Karsai, M., Kivela, M., Pan, R. K., Kaski, K., Kertész, J., Barabási, A.-L., & Saramäki, J. (2011). Small but slow world: How network topology and burstiness slow down spreading. *Physical Review E*, 83(2), 025102.
- [4] Karrer, B., & Newman, M. E. (2011). Stochastic blockmodels and community structure in networks. *Physical Review E*, 83(1), 016107.

Comment - In the response, the authors also referred to the motivation of this study as: “...there is lack of appropriate methodologies for assessing whether and which ‘parameters’ of the STN model $N = (G, \rho)$ fits empirical data.” In my opinion, there have been few attempts to fit the STN model to empirical data simply because the research community notices that it is too far from reality.

If I can borrow an example from static network analysis, it seems like trying to fit Erdos-Renyi random graph model to empirical network data. I have no doubt about the theoretical and historical importance of the ER random graph. However, I never expect the goodness of its fit to empirical networks because real-world networks are probably more heterogeneous and well-structured than the ER random graph. In a similar way as this, the research community respects the theoretical importance of the STN model, but does not regard it as a good model of reality, on the basis of numerous studies about the different kinds of correlations observed in empirical temporal networks and their impacts on diffusion dynamics.

The failure to accurately work on empirical temporal networks makes the inference of the STN model not convincing to general readers as a practical tool.

Response - Thank you for giving this good example. The ER random graph can be regarded as the null model of a network that preserves the edge number only (or, equivalently, the average node degree). Indeed, this is so inadequate as a descriptor of networks in reality. However, a good generative model should flexibly suit different requirements of modelling accuracy, and here, in this somewhat extreme and perhaps degenerate case, the ER random graph definitely provides the best estimate in the sense of maximum entropy, if the only information available is edge number. In fact, the ER graph is the simplest exponential random graph model (ERGM)², a long-standing tool practically used in social network analysis [5–7]. In particular, a vigorous research effort is coupled to use the extended temporal exponential random graph model (TERGM) to model time-varying empirical networks (however, in a rather different way, see Supplementary Note 1 for a brief review).

As the third reviewer commented, our proposed method for reconstructing temporal networks is “of course not optimal,” but hopefully it is “a good first step.” We are still of the view that the STN model will be attractive from various aspects for the purpose of extracting statistical properties of temporal networks, despite that the fast algorithm implementation, at least at the current stage, is only feasible for the first-order STN model under quite limited conditions. We hope that, just like static networks being well modelled by ERGM of tunable complexity, the improved inferential algorithms for higher-order STN models will provide a systematic analysis approach for temporal networks.

- [5] Frank, O., & Strauss, D. (1986). Markov graphs. *Journal of the American Statistical Association*, 81(395), 832-842.
- [6] Robins, G., Pattison, P., Kalish, Y., & Lusher, D. (2007). An introduction to exponential random graph (p^*) models for social networks. *Social Networks*, 29(2), 173-191.
- [7] Lusher, D., Koskinen, J., & Robins, G. (2012). *Exponential random graph models for social networks: Theory, methods, and applications*. Cambridge University Press.
- [8] Hanneke, S., Fu, W., & Xing, E. P. (2010). Discrete temporal models of social networks. *Electronic Journal of Statistics*, 4, 585-605.

Comment - 2. Distinction between public and private information.

Thanks to the examples that the authors provided in the response, now I understand what the authors meant by “public” and “private” information. I recommend to rephrase them by “observable” and “unobservable” information, respectively. In the example of viral marketing, the first-arrival-time data is available only to companies, not public, and is definitely containing private information of customers. It requires no argument that the data of “who buys what at when” is very sensitive and private information. In the example of blogspace, indeed one can say that blog posts are public and interactions between bloggers are private. However, I suggest to consider to use “observable” and “unobservable” to explain these two examples in a consistent way. Although my concern about this point might seem irrelevant to science, I strongly believe that this point is really crucial to clarify the core of the motivation of this study.

Response - Replaced. Thank you for the good suggestion.

²The role of ERGM in the network modelling resembles that of Maxwell-Boltzmann distribution in statistical physics. Put simply, the ERGM postulates an exponential family over the topological configuration space of the networks in accordance with given structural properties, called network descriptors, which are typically in terms of independent functions of graph adjacency matrix. The most widely used descriptors include numbers of edge, triangle or higher-order cliques, node degree sequence, average path length, degree-degree correlation, and that related to the eigenstructure of the graph adjacency matrix. When there is only a single descriptor (edge number), the ERGM then reduces to the ER random graph.

Comment - 3. Simulations on empirical temporal networks.

The proposed method assumes that any node u has finite t_u , i.e., the diffusion reaches all nodes in the network in a finite time. However, in empirical temporal networks, not all of node pairs are reachable through time-respecting paths (e.g., see Table 2 in Starnini et al., Physical Review E 85, 056115 (2012)). Was this assumption approved for the data sets shown in Table 2? If not, how can the discrepancy be justified? In Table 2, it is almost impossible to understand the difference between the four experimental settings that led to BEP1-4. Furthermore, the BEP3 and BEP4 are never mentioned in the main text. The experimental settings should be clearly described.

Response - Yes, this (time-respecting) reachability assumption is verified, and the synthesised diffusion cascades used for our empirical validation contain no infinite t_u . Also, as summarised in Table 1, the experimental settings corresponding to BEP₁–BEP₄ are clarified and further discussed in the revised manuscript (also see Supplementary Table S4).

Table 1: Summary of experimental settings used for BEP₁–BEP₄

Algorithm Input	Empirical DAT	Synthesised DAT	Empirical WTD
BEP ₁	√		
BEP ₂		√	
BEP ₃	√		√
BEP ₄		√	√

Empirical DAT: Diffusion data of the cascades realized on empirical contact sequences $N = \{(u_i, v_i, t_i)\}$

Synthesised DAT: Diffusion data of the cascades simulated on the STN model $N_s = (G, \rho)$ fitted to N

Empirical WTD: Empirical distribution of waiting times read off from N (see our response on page 2)

Reviewer #1 (Remarks to the Author):

The authors carefully addressed the concerns I raise in the last round. Now I understand that the use of the STN model is not for realistic modeling but rather for null modeling. I agree that in this sense the framework proposed in the manuscript is an important first step. Therefore, I agree the present manuscript for publication in Nature Communications.